# High-resolution detection of copy number alterations in single cells with HiScanner

Yifan Zhao [1,2], Lovelace J. Luquette [1], Alexander D. Veit[1], Xiaochen Wang[3], Ruibin Xi [3], Vinayak V. Viswanadham [1], Yuwei Zhang [1], Diane D. Shao[4,5], Christopher A. Walsh [4], Hong Wei Yang[6], Mark D. Johnson [6] ✉ & Peter J. Park [1] ✉

Improvements in single-cell whole-genome sequencing (scWGS) assays have enabled detailed characterization of somatic copy number alterations (CNAs) at the single-cell level. Yet, current computational methods are mostly designed for detecting chromosome-scale changes in cancer samples with low sequencing coverage. Here, we introduce HiScanner (High-resolution Single-Cell Allelic copy Number callER), which combines read depth, B-allele frequency, and haplotype phasing to identify CNAs with high resolution. In simulated data, HiScanner consistently outperforms state-of-the-art methods across various CNA types and sizes. When applied to high-coverage scWGS data from 65 cells across 11 neurotypical human brains, HiScanner shows a superior ability to detect smaller CNAs, uncovering distinct CNA patterns between neurons and oligodendrocytes. We also generated low-coverage scWGS data from 179 cells sampled from the same meningioma patient at two time points. For this serial dataset, integration of CNAs with point mutations revealed evolutionary trajectories of tumor cells. These findings show that HiScanner enables accurate characterization of frequency, clonality, and distribution of CNAs at the single-cell level in both non-neoplastic and neoplastic cells.

Over the past decade, advances in single-cell omics technologies have provided an opportunity to interrogate cellular heterogeneity with unprecedented resolution. Single-cell RNA-sequencing has been the most popular[1,2], but significant progress has also been made for single-cell ATAC-sequencing for profiling open chromatin[3] and single-cell Hi-C for profiling three-dimensional folding[4,5]. Another technology that has advanced is single-cell whole-genome sequencing (scWGS), which can be used to uncover cell-to-cell variation in somatic mutations that may be masked in bulk sequencing.

Various scWGS approaches have been developed for detecting copy number alterations (CNAs), primarily in cancer cells. Examples include degenerate-oligonucleotide-primed PCR (DOP-PCR)[6], direct library preparation (DLP+)[7,8], and the now-defunct 10× Genomics chromium CNV assay[9]. However, the fraction of genome covered by these approaches is low (low breadth-of-coverage) and their main application was to detect clonal patterns of chromosomal arm-sized CNAs. Single-cell whole-genome amplification (scWGA) offers high breadth-of-coverage and could improve spatial resolution, but it has

---

[1]Department of Biomedical Informatics, Harvard Medical School, Boston, MA, USA. [2]Program in Health Sciences & Technology, Harvard Medical School & Massachusetts Institute of Technology, Boston, MA, USA. [3]Department of Probability and Statistics, School of Mathematical Sciences, Peking University, Beijing, China. [4]Division of Genetics and Genomics, Boston Children's Hospital, Boston, MA, USA. [5]Department of Neurology, Boston Children's Hospital, Boston, MA, USA. [6]Department of Neurological Surgery, University of Massachusetts Chan Medical School, Worcester, MA, USA. ✉e-mail: mark.johnson3@umassmemorial.org; peter_park@hms.harvard.edu

been hindered by the noise introduced during amplification. Earlier scWGA methods have involved exponential or quasi-linear amplification, generating highly uneven coverage across the genome, allelic bias (in which one allele is amplified at a higher rate), and frequent allelic dropout (ADO) (where only one allele is amplified). Even if the cell is sequenced at high depth, these amplification artifacts can confound CNA detection by causing large fluctuations in the read depth ratio (RDR; ratio between observed and expected read count), yielding relatively few advantages over low-coverage (<1×, often <0.1×) scWGS.

Recent advances in scWGA chemistry[10,11] and declining sequencing costs have made high-coverage (>20×) scWGS more attractive for CNA detection. In particular, scWGA by primary template-directed amplification (PTA) achieves more uniform amplification across the genome[10,12]. We have already used the high-coverage PTA data to identify somatic single-nucleotide variants in the brain, tracing cellular lineage across different tissues[13], providing a more precise estimate of age-associated mutational rate[12], and characterizing the regions more prone to mutagenesis in different cells[14]. PTA data also offer a potential to improve the resolution of CNA detection beyond the usual Mb-scale[15,16].

The use of B-allele frequency (BAF, the relative proportions of reads covering the two alleles at heterozygous germline single-nucleotide polymorphisms, gHETs) is a standard feature in best bulk CNA methods such as ASCAT[17] and Battenberg[18] for inferring allele-specific copy number state. Two recently developed single-cell callers, CHISEL[19] and Alleloscope[20], leverage phasing and BAF for identifying allele-specific CNAs in low-coverage scWGS. However, Alleloscope performs segmentation on bulk samples, not single cells, limiting its ability to detect rare CNAs, which are critical to understanding the genomic architecture of non-cancerous cells and especially non-dividing cells. CHISEL is not tuned for the discovery of small CNAs (e.g., <5 Mb), potentially obscuring certain biological processes. SCOVAL[21] also utilizes BAF information, but it is primarily a validation approach and not a CNA caller. Current single-cell CNA callers are tailored to the analysis of clonal CNAs in low-coverage cancer cells[19,22–24] and are not designed to fully extract information available in high-coverage scWGS data.

In this work, we propose HiScanner (High-resolution Single-Cell Allelic copy Number callER) for accurate, high-resolution allele-specific CNA detection in scWGS. HiScanner leverages reference-based phasing and local sequence bias to efficiently estimate BAF to complement read depth-based CNA identification. Building upon our previously developed bulk CNA caller framework[25], HiScanner performs read depth normalization at nucleotide resolution, followed by joint segmentation of single-cell profiles using the Bayesian information criterion (BIC). By simultaneously modeling RDR and BAF for individual cells, HiScanner infers cell ploidy and integer allele-specific copy number states. It also considers assay-specific ADO patterns to find an optimal bin size, a nuanced yet crucial parameter in CNA analysis. We demonstrate that HiScanner outperforms four representative single-cell CNA methods—CHISEL (BAF-aware), Ginkgo, SCOPE, and SCYN (all BAF-agnostic methods)[19,22–24]—in real and simulated data with greater sensitivity and precision at small size scales (200 kb–5 Mb). For biological discovery, we analyzed two datasets. Application of HiScanner to a high-coverage scWGS dataset from eleven neurotypical human brains uncovered a large number of previously undetected small CNAs, revealing contrasting patterns between neuronal and glial cells. We also sequenced 179 cells at low coverage (~0.5× on average) from biopsies of an initial ($n = 88$) and recurrent ($n = 91$) meningioma tumor pair. HiScanner revealed phases and timing of tumor progression by detecting subclone-defining CNAs, which were subsequently corroborated by an orthogonal clustering based on somatic point mutations[26]. Lastly, we introduce ViScanner, a web-based visualization tool built using the HiGlass architecture[27], which offers users the ability to navigate genomic data with continuous zooming, allowing for user-friendly exploration of genomic regions at various resolutions.

## Results

### Overview of the algorithm

HiScanner introduces three key improvements over existing single-cell CNA detection methodologies to enhance the resolution of CNA discovery (Fig. 1). First, optimal resolution is inferred by determining the smallest permissible bin size based on the size of amplification artifacts, rather than leaving the tuning of this critical parameter to the user. Second, joint segmentation is performed across cells to avoid overfitting amplification noise. Third, an efficient BAF estimator is implemented to mitigate the effects of phase-switch errors and ADO. In addition to these features, HiScanner corrects for bias in read depth due to local GC content using a nucleotide-level model of GC bias[25]. These improvements are described in more detail below.

**Selecting optimal bin size based on allelic dropout events.** Variations in RDR across the genome are the primary signal used to detect copy number changes, but scWGA introduces considerable levels of uneven amplification along the genome that masquerade as CNAs. A prevalent form of uneven amplification is ADO, where failed amplification of one allele results in an RDR decrease similar to a copy number loss (Supplementary Fig. 1a). Previous single-cell CNA callers partitioned the genome into large bins to dampen the effect of segments lost to amplification, but this limits the size of CNAs that can be detected. On the other hand, decreasing the bin size unmasks RDR fluctuations, possibly inducing false-positive CNA calls. A key insight exploited by HiScanner is that the size distribution of ADO events, which is usually bounded above and is specific to assay-dependent amplicon length, can be used to determine a balanced bin size that maximizes detection resolution and minimizes the effects of amplification noise (Supplementary Fig. 1b). Furthermore, because dropout sizes and frequencies can differ considerably between scWGA protocols[28] (Supplementary Fig. 1c), it is important to fit the dropout size distribution on each dataset. For example, the average size of dropout events in two cells amplified by PTA and MDA[29] differed by two orders of magnitude ($10^3$ bp versus $10^5$ bp, respectively, Supplementary Fig. 1d). HiScanner employs a two-state hidden Markov model (HMM) to detect dropout events using BAF at gHETs (Supplementary Fig. 1b). Subsequently, the predicted dropout events are used to build a sample-specific dropout size distribution and the top 5th percentile of dropout sizes is used as the smallest permissible bin size.

**Joint segmentation across multiple cells.** By leveraging signals from multiple cells, HiScanner's joint segmentation promotes breakpoint sharing, thus improving detection sensitivity. We extended our previous strategy[25] of merging adjacent bins with similar copy number state (as determined by the BIC) to the multi-sample case by computing a BIC from the joint breakpoint likelihood of a breakpoint across cells (Methods). This merging approach retains information about bin position, avoids overfitting by penalizing excessive segmentations, and amplifies signals of clonal CNAs. Notably, the clustering approach proposed in CHISEL to simultaneously model BAF and RDR across all cells does not consider the proximity of bins, making the method more susceptible to calling false-positive breakpoints, especially in regions with complex structural variations or relatively high amplification noise.

**Accurate estimation of the B-allele frequency (BAF).** BAF can provide additional information for CNA inference where RDR is ambiguous; a shift in the BAF from the expected value of 0.5 in diploid regions suggests an underlying CNA (for example, a single copy gain shifts the BAF to 1/3 or 2/3, depending on which allele the BAF is

measured against). In bulk data, obtaining allele-specific read depth at gHET loci is an effective way to compute BAF; however, in scWGS, uneven amplification between homologous alleles (allelic bias) and dropout can hinder this approach. Furthermore, low-coverage sequencing—which is more typical for single-cell CNA studies—reduces the number of reads at gHETs, thereby increasing BAF noise with its undersampling. The effects of both allelic bias and gHET undersampling on BAF estimation can be mitigated by summing allele-specific read depth over bins. However, this requires phasing each gHET to a haplotype, and phase switch errors, which can occur at rates of 1–2%[30,31] for reference-based phasing approaches, confound bin-level BAF estimation by assigning depth to the incorrect allele. CHISEL proposed a two-step process to calculate per-bin BAF estimates: after an initial phasing based on an external reference panel[32,33], expectation-maximization (EM) is used for refinement, first aggregating phased gHETs into haplotype blocks and then performing EM refinement of phase across cells. Alleloscope also adopted an EM approach, but it operates on unphased, individual gHETs. Although effective, the EM step used by both CHISEL and Alleloscope is computationally expensive and is challenging to apply to high-resolution

bins. Indeed, CHISEL operates on ~5 Mb bins (by default), and Alleloscope only computes BAF on the final copy number segments derived from bulk or pseudobulk.

HiScanner computes BAF by phasing gHETs using a reference-based method followed by a computationally efficient, two-step procedure to reduce the effect of phase-switch errors and dropout artifacts (Supplementary Fig. 2; Supplementary Note). In the first step, LOH is detected by exploiting GC bias, a source of non-biological fluctuations in total read depth that is treated solely as a nuisance. In a segment without LOH, GC-induced read depth changes—which are roughly evenly split between homologous alleles due to similar nucleotide content—lead to a subtle but detectable difference in the distributions of BAF before and after a haplotype-specific *shift* operation (Supplementary Note). In regions affected by LOH, this shift procedure has little effect. Thus, HiScanner predicts the presence of LOH in each segment by a Kolmogorov-Smirnov test of the BAF and shifted BAF distributions. In the second step, peaks (local maximums) are called in the per-bin BAF distribution of each segment. If the segment was not predicted to contain LOH, then any BAF peak near 0 or 1 is interpreted as dropout artifacts and removed, and

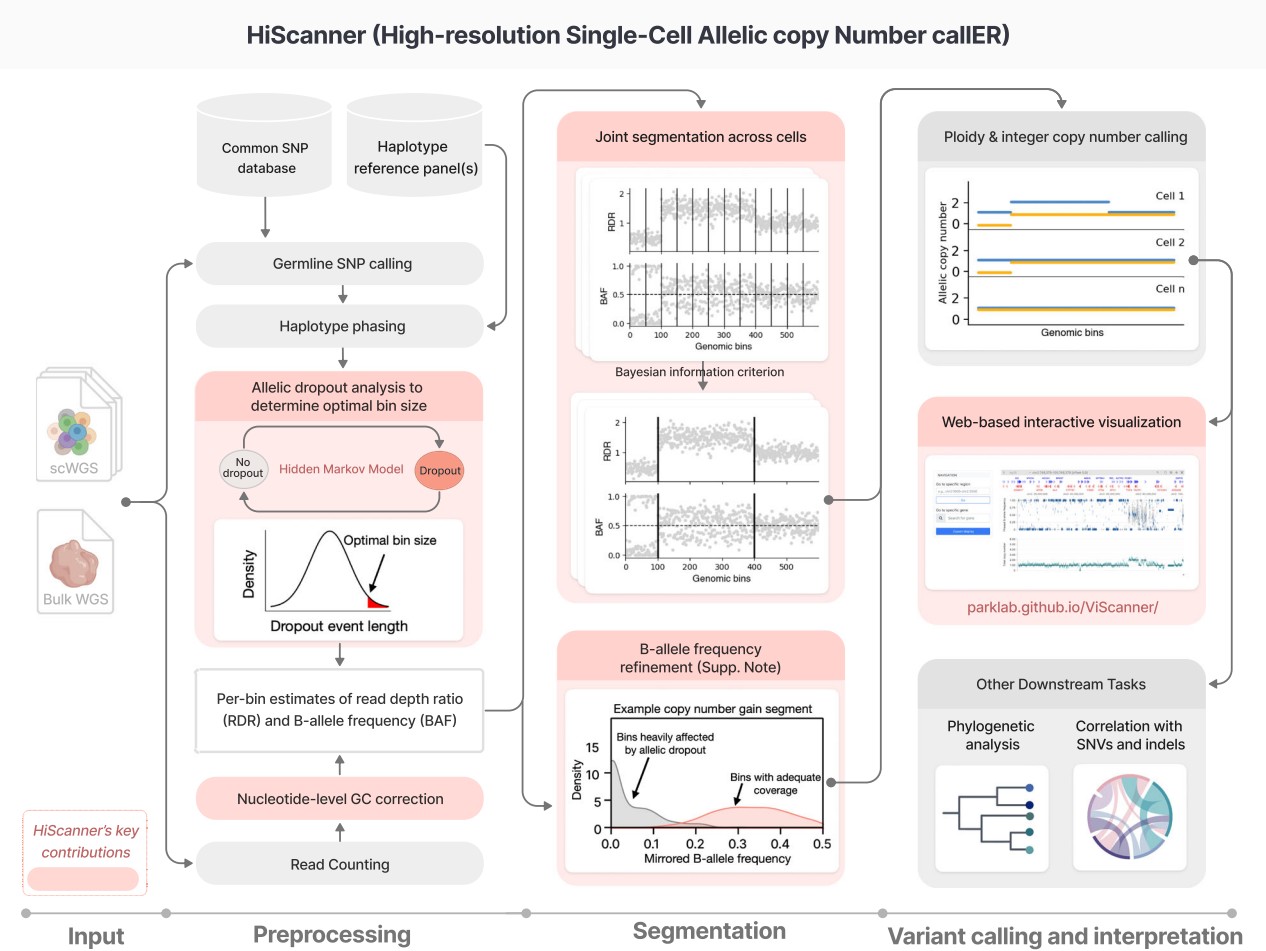

**Fig. 1 | HiScanner algorithm overview.** HiScanner starts with scWGS data as input. In the preprocessing step, gHETs are called from bulk and phased using population-level reference panels. Normalization is performed at the nucleotide level to remove GC bias from read depth signals. Subsequently, a two-state hidden Markov model is employed to estimate ADO event lengths, which are then used to find the optimal bin size by setting a cutoff at the top 5th percentile of the length distribution. For each cell, HiScanner aggregates allele- and haplotype-specific read counts per bin to compute BAF and performs semi-parametric regression to compute RDR. In the segmentation step, HiScanner jointly partitions the genomes across all cells using BIC to identify breakpoints. An LOH test is then used to further

refine the BAF estimates within each segment (see Supplementary Note for details). In the variant calling and interpretation step, for each cell, HiScanner simultaneously infers the cell-specific scale factor and allelic copy number for each segment by modeling the RDR and BAF as Gaussian distributions. A web-based visualization tool, ViScanner, is also provided, which enables interactive visualization of HiScanner CNA calls at any resolution. Other downstream analyses of HiScanner output include tasks such as phylogenetic inference and correlation of CNAs with SNVs and indels. HiScanner's key contributions are highlighted in light red boxes. Created in BioRender. Chun, E. (2025) https://BioRender.com/bpgwigx.

**Table 1 | Qualitative comparison of five single-cell copy number callers (using default parameters)**

| | Input signal type (s) | GC correction | Bin size (kb) | Segmentation algorithm | Joint segmentation across cells | Integration of phasing and BAF |
|---|---|---|---|---|---|---|
| Ginkgo | RDR | Bin-level | 500 | CBS | No | N/A |
| SCOPE | RDR | Bin-level | 500 | CBS | Yes | N/A |
| SCYN | RDR | Bin-level | 500 | BIC-based | Yes | N/A |
| SCOVAL | RDR, BAF | Bin-level | 500 | CBS | Yes | Filtering step |
| HiScanner | RDR, BAF | Nucleotide-level | 100 | BIC-based | Yes | CNA calling step |
| CHISEL | RDR, BAF | Not performed | 50,000 | K-means clustering | Yes | Segmentation step |
| Alleloscope | RDR, BAF | Not performed | N/A[a] | HMM | No | CNA calling step |

*RDR* read-depth ratio, *BAF* B-allele frequency, *CBS* circular binary segmentation, *BIC* Bayesian information criterion, *HMM* hidden Markov model.
[a]Alleloscope models gHETs directly and does not perform binning.

the maximum of the remaining non-zero peak is the final segment BAF. Otherwise, the BAF is set to 0. The efficiency of this procedure is critical for CNA analysis with high-resolution bins.

**Visualization using ViScanner.** To enhance the accessibility of HiScanner output, we developed a web-based visualization platform, ViScanner (Methods). Traditional single-cell CNA analysis tools often generate static output, including mapping coordinates for potential CNAs and genome-wide visualizations of CNAs. However, genome-scale plots do not adequately support the exploration of small CNAs. We developed an interactive React-based web application, ViScanner, which leverages the HiGlass toolkit[27] to provide a user-friendly, dynamic resolution CNA browser. ViScanner allows users to scrutinize HiScanner output with real-time zooming, akin to the user experience in web-based services like Google Maps. Importantly, ViScanner ensures data security by executing JavaScript locally in the user's browser, avoiding the need to upload data, even temporarily, to any external server.

## Comparison with other methods

To assess HiScanner's performance, we conducted benchmarking studies using both real and simulated data. HiScanner was compared against four widely-used single-cell CNA methods, Ginkgo[22], SCOPE[23], SCYN[24], and CHISEL[19]. Ginkgo is a popular web-based application that identifies CNAs based on RDR and performs segmentation via circular binary segmentation[34]. SCOPE is a normalization and copy number estimation method that uses a Poisson latent factor model to account for technical biases and copy number changes; SCYN employs SCOPE's normalization and employs dynamic programming for efficient copy number segmentation. CHISEL is a more sophisticated algorithm that combines phasing, BAF, and RDR to identify allele-specific CNAs. Our benchmarking analysis focused on the detection of non-clonal CNAs (private to a single cell), which are more difficult to identify than clonal CNAs (present in a subpopulation). Consequently, methods designed to detect only clonal events, such as Alleloscope[20], were excluded to avoid mischaracterizing their performance (Table 1).

We first evaluated the three callers on detecting nonclonal CNAs using high-coverage simulated data. Previous benchmarking studies often simulated single-cell CNAs by manipulating read counts within pre-determined CNA locations, thereby focusing exclusively on RDR-based CNA callers[35,36]. However, allele-specific callers cannot be properly assessed by this strategy because simulation at the level of read counts does not preserve the haplotype structure. Moreover, although simulating read counts using Lorenz curve sampling[35] can recapitulate read count variability associated with GC content, it does not faithfully capture sampling and amplification noise characteristic of scWGS sequencing data. To overcome these limitations, we adapted the synthetic diploid X (SDX) chromosome methodology[11] to simulate somatic CNAs. In this approach, sequencing reads sampled from two male haploid X chromosomes from different individuals are pooled to generate a diploid chromosome (Fig. 2a). By sampling reads prior to pooling, it is possible to insert copy gain or loss events specific to a single allele without disrupting local haplotype structure. Further, one can explicitly control phasing accuracy and phase switch rates, while preserving scWGS amplification artifacts.

We simulated four types of non-clonal CNAs with sizes ranging from 200 kb to 5 Mb using the SDX: single-copy losses, single-copy gains, two-copy gains, and copy-neutral loss-of-heterozygosity (cnLOHs). Our analysis of four CNA types (200 kb–5 Mb) revealed distinct performance patterns across tools. For single-copy losses (Fig. 2b,c), HiScanner and Ginkgo excelled at 1 Mb and above, with HiScanner achieving slightly better sensitivity (~0.90 vs 0.85) while matching Ginkgo's high precision (~0.96). CHISEL showed moderate performance (sensitivity 0.80, precision 0.51), while SCOPE and SCYN demonstrated limited sensitivity despite reasonable precision. In single-copy gain detection (Fig. 2d, e), HiScanner outperformed others at smaller sizes (~0.65 sensitivity at 500 kb), with Ginkgo following a similar but lower trajectory. Both achieved strong performance (>0.90) above 2 Mb, while CHISEL showed moderate results. SCOPE and SCYN performed poorly across all sizes.

For two-copy gains (Fig. 2f, g), performance differences narrowed above 1 Mb, with HiScanner and Ginkgo showing nearly identical high performance (~0.85 sensitivity, ~0.90 precision at 2.5 Mb+). CHISEL matched their precision for larger events but lagged in sensitivity. SCOPE and SCYN completely missed these events - after inspecting SCOPE's normalized read depths (SCYN uses the same normalized profile as SCOPE), we found that SCOPE's normalization algorithm interprets these amplifications as technical errors and over-corrects for them. While this cautious approach makes sense when analyzing hundreds of cells, it becomes problematic with just a few cells since distinguishing real biological signal from technical artifacts becomes more challenging.

In cnLOH detection (Fig. 2h, i), only HiScanner and CHISEL showed capability, with CHISEL demonstrating sharp improvement around 1 Mb (>0.90 precision and sensitivity for larger events), while HiScanner showed gradual improvement. CHISEL's increased sensitivity for large cnLOH events might be attributed to its joint clustering approach, which utilizes both RDR and BAF, whereas HiScanner's multi-sample segmentation is based only on RDR. The other tools were not designed for cnLOH detection due to their BAF-agnostic nature (Table 1). Overall, HiScanner demonstrated the most consistent performance across CNA types, while other tools showed variable effectiveness depending on the variant type and size, with all tools struggling with CNAs below 500 kb.

Figure 2j illustrates a representative ~1 Mb single copy loss. In this example, while all five methods detected a change in the region, only HiScanner, Ginkgo, and SCOPE made a precise call matching the ground truth boundaries. CHISEL correctly identified the loss but predicted spurious copy number gains in the flanking regions, likely due to sensitivity to coverage fluctuations. SCYN showed multiple

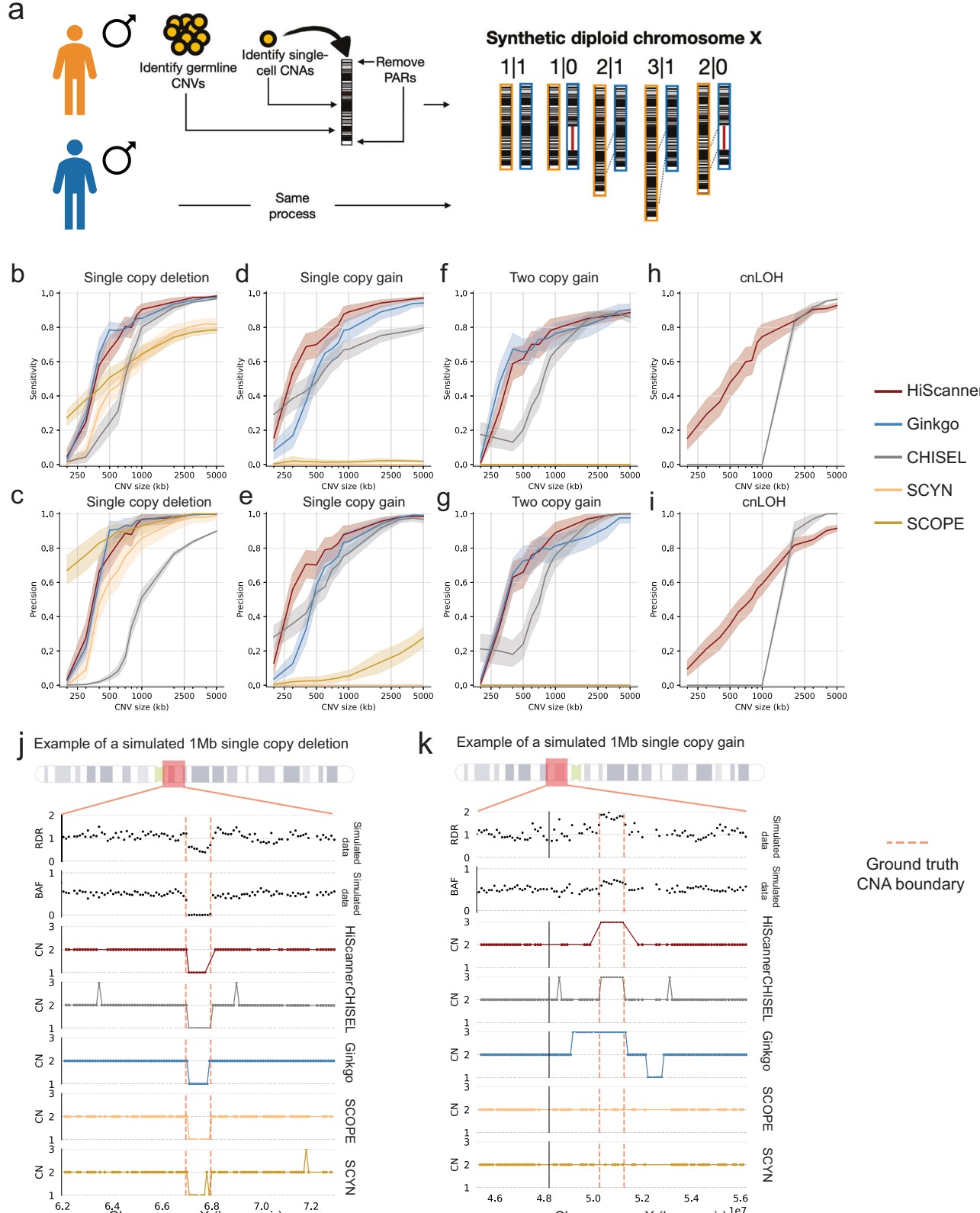

**Fig. 2 | Method comparisons on simulated data. a** Construction of synthetic diploid X chromosomes. First, two single cells are obtained from different male donors. Next, reads are extracted from chromosome X, excluding reads from pseudoautosomal regions, and germline or somatic CNAs identified in the original cells are removed. This serves as a template for creating various non-clonal CNA conditions, including single copy loss (1|0), single copy gain (2|1), two copy gain (3|1), and cnLOH (2|0), after which the reads are merged to create a BAM file for each mutated synthetic diploid chromosome X. **b**–**i** Evaluations of precision and sensitivity for five callers for simulated single copy losses, single copy gains, two copy gains, and cnLOHs. Shaded regions indicate 95% confidence interval bands calculated from bootstrap sampling. Examples illustrating caller discrepancies in detecting a simulated single copy loss (**j**) and a simulated single copy gain (**k**). Ground truth CNA boundaries are shown as red dashed lines.

errors, including a spurious loss within the true CNA region and an incorrect gain outside of the region.

For the single copy gain (Fig. 2k), the event is characterized by an increase in RDR and shift in BAF. HiScanner made a clean detection of the gain, while CHISEL and Ginkgo both identified the event but showed incorrect copy number states in the surrounding regions. SCOPE and SCYN failed to detect the gain, maintaining a baseline copy number state throughout the region. These examples illustrate HiScanner's superior ability to make precise calls while avoiding false positives due to technical noise in both RDR and BAF signals.

We next assessed the impact of bin size and sequencing depth on HiScanner's performance across different CNA types. For bin size analysis, we compared 50 KB, 100 KB, and 500 KB bins in simulated 30× data (Supplementary Fig. 3a). While smaller bins (50 KB) showed improved sensitivity for detecting variants below 500 kb, they generally exhibited lower precision due to increased noise. The 100KB bin size provided optimal balance between sensitivity and precision across most CNA types, particularly for variants >1 Mb. Larger bins (500 KB) demonstrated reduced performance, especially for smaller variants. Regarding sequencing depth, we evaluated 0.5×, 10×, and 30× coverage (Supplementary Fig. 3b). Higher sequencing depth (30×) consistently improved both precision and sensitivity across all CNA types, though the performance gain between 10× and 30× was modest. Low-coverage sequencing (0.5×) showed substantially lower performance, particularly for single-copy events and smaller variants, though it maintained reasonable accuracy for larger CNAs (>2 Mb) and two-copy gains. Notably, the performance difference between 10× and 30× coverage was less pronounced for cnLOH events, suggesting that moderate coverage might be sufficient for detecting these variants. These results validate that the default parameter used in our benchmarking experiment (30× depth, 100 KB bins) as providing optimal performance while indicating that cost-effective alternatives (e.g., 10× coverage) might be suitable for specific applications.

As an illustrative example using real data, we next applied the five methods to detect CNAs in a single human neuron amplified by PTA and sequenced to 29× (Fig. 3a). When examining the genome-wide copy number profiles, HiScanner and Ginkgo showed largely consistent patterns in detecting major CNAs across chromosomes, revealing distinct loss events in chromosomes 2, 5, 6, 8, 11, 12, and 17. These events span various sizes from 300 kb to 170 Mb (whole-chromosome), with some potentially biallelic losses. SCYN and SCOPE demonstrated similar capabilities in identifying these major events, though with slightly different segmentation patterns. In contrast, CHISEL produced a notably different profile, suggesting an overall tetraploid state with more fragmented segmentation, even when using 100 kb bins. After parameter adjustment with 5 Mb bins, CHISEL showed improved stability but still maintained higher baseline copy numbers compared to other methods.

The differences among these methods became more apparent when examining their behavior in specific genomic contexts (Fig. 3b). Near centromeric regions, Ginkgo and CHISEL frequently reported focal amplifications with high copy numbers, suggesting insufficient correction for the highly repetitive sequences in these regions. HiScanner, however, maintained more stable signals across these regions, likely due to its use of uniquely mappable positions and a nucleotide-level normalization approach. SCYN and SCOPE also showed robust performance in centromeric regions, avoiding the spurious high copy number calls seen with other methods. The varying complexity of CNA calls among methods is quantified in Fig. 3c, where CHISEL showed the highest rate of segmentation (10,322 breakpoints, or 0.38 per 100 kb). In comparison, HiScanner, Ginkgo, SCYN, and SCOPE maintained more conservative segmentation patterns, suggesting a better balance between sensitivity and specificity in CNA detection. When examining the size distribution of detected CNAs and their BAF support (Fig. 3d), we observed that Ginkgo tended to call small copy number loss events

that lacked corresponding BAF evidence. HiScanner showed better concordance between copy number calls and BAF signals, particularly for larger events.

To further demonstrate the performance differences among these methods, we examined two specific genomic regions in detail: a diploid region with uneven read coverage (Fig. 3e) and a single copy loss candidate (Fig. 3f). In both cases, HiScanner maintained stable copy number calls despite coverage fluctuations, while the other methods showed more variable responses to these coverage differences.

In summary, our analysis of both simulated data and an aneuploid single neuron revealed distinct characteristics of each caller. Ginkgo displayed high sensitivity to changes in RDR but, without BAF information, lacked precision in detecting small CNAs and was not capable of cnLOH detection. CHISEL tended to overfit when used with small initial bin sizes and, in a single-sample context, produced an improbably large number of breakpoints. Given that CHISEL is optimized for megabase-scale clonal CNA detection (e.g., in cancer samples), this behavior is expected. SCYN showed limited sensitivity across different CNA types in simulated data, and while it could identify major copy number events in real data, it sometimes produced spurious calls in regions with coverage fluctuations. SCOPE demonstrated more conservative calling patterns, maintaining reasonable precision but with reduced sensitivity, particularly for smaller events and subtle copy number changes. Overall, HiScanner maintained a balance between sensitivity and precision in detecting CNAs at high resolution.

## Cell-type-specific CNA patterns in neurotypical human brains

Previous studies using scWGS to investigate non-diseased human brains have reported contradictory findings regarding the frequency of CNAs in neurons. A meta-analysis of multiple low-coverage scWGS reported a linear decline of CNA neuron frequency with age ($R^2 = 0.92$, range 0–40.0%) in brains aged 0.36–95 y.o[15]. Knouse et al. reported a significantly lower frequency (2.2%; 95% CI 0.3–7.9%) for large CNAs in human brain cells[37]. More recently, Kalef-Ezra et al. applied PTA to human brain cells and found 33.8% to have at least one megabase-scale CNA[12,14]. These discrepancies highlight the need for further research into the prevalence of CNAs in the human brain.

Using HiScanner, we analyzed previously published high-coverage PTA scWGS data encompassing 65 single neurons and oligodendrocytes from eleven neurotypical human brains spanning an age range from 0.4 to 83 years[12,14] (Fig. 4a; Supplementary Table 1). HiScanner was applied separately to each brain, as clonal CNAs should be specific to each individual. Large-scale aneuploidy, characterized by multiple chromosome-arm level losses, was clearly observed in two cells out of 65 (3.1%). These numbers are concordant with Knouse et al.[37] than the others.

Beyond these instances, HiScanner revealed an additional 614 smaller, mostly private CNAs (314 private losses and 57 private gains) in the rest of the cells. Whereas previous approaches were limited to a lower size bound of ~3 Mb on somatic CNA detection in single cells[15,16,36], nearly all (609/614, 99.3%) of the CNAs identified by HiScanner were smaller than 3 Mb (Fig. 4b, c), with a median size of 612 Kb (range 300 Kb–24.1 Mb). Using the size-specific sensitivities and specificities from our simulated data, we estimated a false discovery rate of 26.3% among the 614 calls (Methods; Supplementary Table 2). In total, 77% (24/31) of neurons and 53% (17/32) of oligodendrocytes contained at least one CNA. We identified 72 clonal events (defined as CNAs sharing identical start and end bin coordinates across multiple cells), with the majority (50/72) being shared between more than two cells, demonstrating substantial clonal diversity. Analysis of age-related patterns using linear mixed effects (LME) models revealed a weak negative correlation between age and non-clonal CNA gains in oligodendrocytes (Supplementary Fig. 4a; slope = −0.020 CNA/year, $p = 0.010$). No significant age-related patterns were observed for

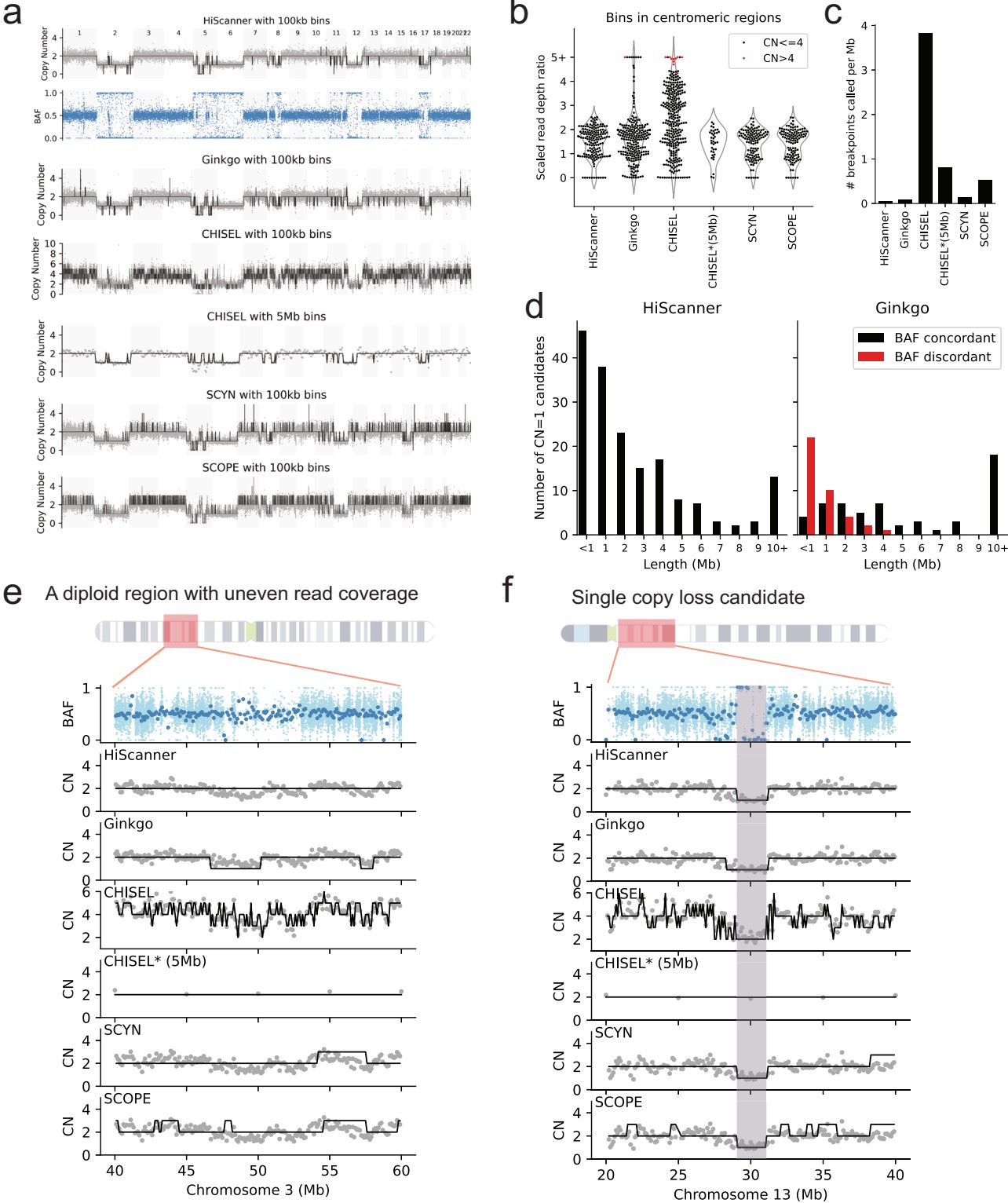

**Fig. 3 | Method comparison in a single, aneuploid PTA neuron. a** BAF track of single neuron 5823PFC-B using 100 kb mappable bins, and corresponding RDR tracks overlaid with total copy number calls reported by different methods: HiScanner, Ginkgo, CHISEL, and CHISEL after parameter optimization on 5 Mb bins, SCYN, SCOPE. **b** RDR distribution for bins inside centromeric regions. CHISEL*: CHISEL on 5 Mb bins. **c** Breakpoint density of each method's call set. **d** Length distribution of BAF-concordant or BAF-discordant loss events called by HiScanner and Ginkgo. All HiScanner calls were BAF-concordant (i.e., y value = 0 for the red bars). **e** An example of a diploid region showing uneven read coverage. **f** An example showing a 1 Mb loss on chromosome 13. The shaded region indicates the location of a CNA candidate.

neuronal gains (Supplementary Fig. 4a; slope = −0.015 CNA/year, $p = 0.218$) or for non-clonal losses in either cell type (Supplementary Fig. 4b; neurons: slope = 0.003 CNA/year, $p = 0.976$; oligodendrocytes: slope = 0.002, $p = 0.589$), suggesting that the relationship between

CNAs and aging may be more complex than described in prior literature[15].

Beyond these temporal patterns, our analysis revealed striking differences in CNA profiles between cell types. Neurons contained

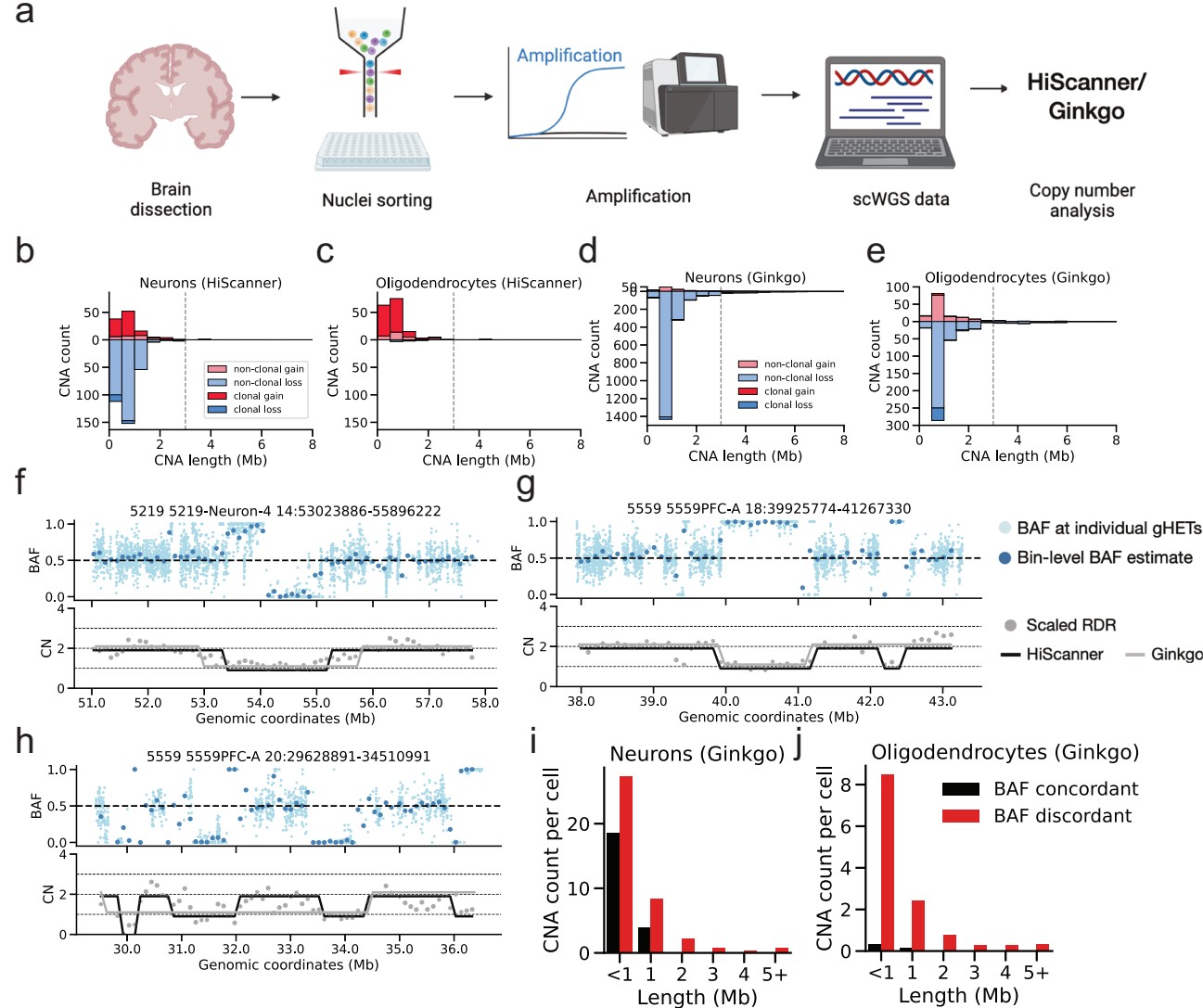

**Fig. 4 | Application to somatic mosaicism in human brain cells. a** Experimental workflow. Nuclei from neurons and oligodendrocytes were isolated from 11 human brains, followed by whole-genome amplification, sequencing, and somatic CNA detection using Ginkgo and HiScanner. Created in BioRender. Chun, E. (2025) https://BioRender.com/hk8kcyi. Length distribution of HiScanner CNA calls, categorized by type and clonality, in 31 neurons (**b**) and 32 oligodendrocytes (**c**). **d, e** Length distribution of Ginkgo CNA calls. **f** Example of a diploid region with uneven coverage where a spurious loss is called by Ginkgo. **g** Example of two adjacent single copy losses, one of which is missed by Ginkgo. **h** Example of a 2 Mb single copy loss. Shaded regions indicate positions where Ginkgo calls are discrepant with HiScanner and BAF. **i, j** Number of Ginkgo calls that are concordant or discordant with BAF, categorized by length and cell types.

both gains and losses, with losses being more prevalent, whereas nearly all CNAs in oligodendrocytes were gains (Fig. 4b, c). In comparison to oligodendrocytes, the odds ratio of observing losses compared to gains in neurons is 61 (*p*-value: 1e − 23, Fisher's Exact Test). Classification into clonal vs non-clonal categories revealed broad distinctions: only 5% of the losses in neurons are clonal, whereas 78% of neuronal gains are clonal. Since neurons are typically non-proliferative, this observation may suggest differential timings of gain and loss events: gains, which were mostly shared, must have occurred early in development prior to terminal differentiation into a post-mitotic neuron, whereas losses, which were mostly private, must have occurred after development in the post-mitotic state. In oligodendrocytes, the majority of CNAs are clonal gains (81% of gains being clonal and no losses), similar to the high clonal fractions in neuronal gains. The differences between neuronal and oligodendrocyte CNAs suggest differential timing and/or mutagenic processes.

For cnLOHs, we focused our analysis on events larger than 2 Mb where our simulations demonstrated high precision and sensitivity (-0.9). This analysis identified cnLOH events in 8 neurons, with one neuron (5817PFC-B) showing an extensive pattern of 12 distinct cnLOH events while the remaining neurons harbored 1–3 events each. Two of these cnLOH events were clonal, appearing in pairs of neurons from the same individual, while the rest were private. Notably, no cnLOH events were detected in oligodendrocytes, similar to the cell-type specific pattern observed with single copy losses.

Analysis with Ginkgo yielded a fundamentally different view of neuronal and oligodendrocyte CNAs, even though a similar range of CNAs were detected (Fig. 4d, e). Notably, Ginkgo called nearly a 10-fold more private neuronal losses than HiScanner and a large number of private losses in oligodendrocytes, whereas HiScanner detected almost none. We assessed the quality of these calls by inspection of the BAF distribution in the putative CNAs and found that a considerable proportion of Ginkgo's calls disagreed with local BAF signals (Fig. 4f–j): 97% (581/599) of CNAs called in oligodendrocytes and 66% (1410/2121) of CNAs called in neurons were BAF-discordant. This suggests that Ginkgo's calls in PTA scWGS data should be filtered based on CNA size

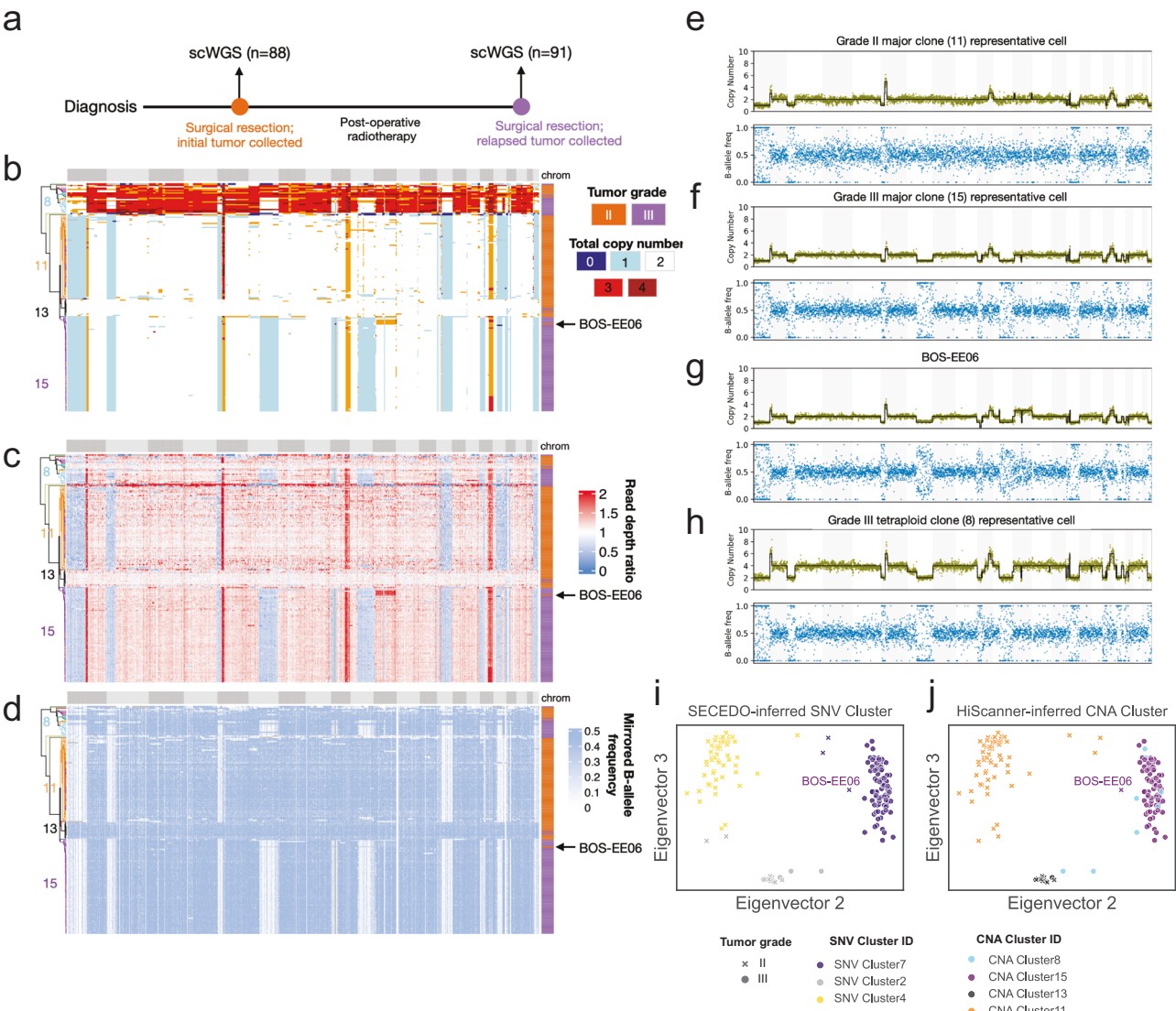

**Fig. 5 | Application to low-coverage scWGS of a recurrent meningioma tumor pair. a** 179 single cells from a paired initial and recurrent meningioma tumor sample were amplified by PTA and sequenced. Total copy number (**b**), RDR (**c**), and BAF (**d**) matrices output by HiScanner. Cell ordering is determined by hierarchical clustering applied to the copy number matrix. Each row represents one cell and each column corresponds to one genomic bin (500 kb mappable). Total copy number and BAF profiles of four representative cells from the major clone of tumor grade II (**e**), major clone of tumor grade III (**f**) grade III, BOS-EE06 (**g**), and tetraploid clone from tumor grade III (**h**). Scatter plots of the second and third eigenvectors of the Laplacian similarity matrix output by SECEDO, colored by SECEDO-inferred SNV clusters (**i**) and HiScanner-based CNA hierarchical clusters (**j**).

or be subjected to BAF-based post-call filtering strategies[21] to ensure the reliability of small CNA calls. In contrast, HiScanner's integration of BAF and RDR signals was effective in preventing these types of errors.

## Integrative CNA and SNV analysis in longitudinal meningioma scWGS

To demonstrate the applicability of HiScanner to low-coverage scWGS, we sequenced PTA-amplified single cells from meningiomas, the most common primary intracranial tumor, to an average depth of 0.5× (Supplementary Table 3). High-grade (grade II–III) meningiomas are notorious for their high recurrence rate despite treatment, and the effectiveness of postoperative radiotherapy in managing these recurrences is still a subject of debate[38,39]. Characterizing the timing of oncogenic events and their evolution in meningiomas may therefore prove helpful in assessing the impact of therapeutic interventions and guiding clinical management strategies. We obtained tumor tissue samples from initial and recurrent (grade II/III) meningiomas from a single patient. 88 and 91 single tumor cells were sorted from the grade

II and grade III specimens, respectively, totaling 179 cells (Fig. 5a, Methods). All cells were amplified by PTA, sequenced to 0.5×, and analyzed for CNAs by HiScanner.

Hierarchical clustering on the copy number profiles from both initial (grade II) and recurrent (grade III) tumors revealed four main clusters (Fig. 5b–d). Cluster 13, which is a mixture of grade II/III cells, consists of cells that lack any large CNAs. Cluster 11 is entirely composed of grade II cells and exhibits hallmarks of high-grade meningiomas such as single-copy losses in chromosomal regions 1p and 22q (Fig. 5e). Cluster 15 is dominated by grade III cells which inherited all CNAs present in Cluster 11 along with additional loss events on chromosomes 6q, 9p, and 11q (Fig. 5f). Interestingly, one grade II cell (BOS-EE06, Fig. 5g) was assigned to Cluster 15, suggesting that it may be closely related to the origin of a major grade III clone. If true, this would imply that the additional copy number events found in grade III tumor cells likely occurred in the natural course of tumor evolution rather than as a consequence of adjuvant therapy. Lastly, Cluster 8 represents a small, distinct group of tetraploid cells from the grade III tumor

(Fig. 5h), exhibiting a range of copy number states from two to six, including odd copy numbers. Tetraploidy hints at a potential tumor progression pattern involving whole-genome duplication, which is rarely reported in the meningioma literature; however, further experiments (e.g., fluorescence in situ hybridization) would be essential to validate this ploidy estimate. The heterogeneity of these clusters reflects the complex nature of meningioma evolution and underscores the importance of single-cell profiling in delineating tumor progression.

Next, we investigated whether cluster assignments derived from CNAs were consistent with patterns of somatic SNVs (sSNVs). To this end, we employed a recently developed somatic SNV-based cell clustering method, SECEDO[26]. Initially, applying SECEDO with its default settings failed to differentiate between aneuploid and diploid cells, which are a hallmark of meningiomas. However, upon refining the input by limiting to sSNV locations identified in matched bulk WGS via GATK MuTect2[40] (Methods), SECEDO yielded reasonable clustering results. SECEDO predicted three primary cell clusters, which closely corresponded to HiScanner's CNA-based clustering (Fig. 5i, j): SNV Cluster 4 aligned largely with CNA Cluster 11; sSNV Cluster 2 comprised a mix of cells from CNA Clusters 8 and 13; and sSNV Cluster 4 was found to be a mix of CNA Clusters 8 and 15. Notably, however, SECEDO's clustering did not distinguish diploid and tetraploid grade III cells. A possible explanation for this is that the whole-genome duplication preceded the majority of sSNV events. SECEDO also assigned three grade II tumor cells to its grade III major cluster, suggesting that some sSNVs preceded CNAs in the evolution of this tumor. To quantitatively compare the sSNV and CNA cluster assignments, we computed the Adjusted Rand Index (ARI), a score ranging from 0 (random labeling) to 1 (perfect match). The two orthogonal clusterings produced an ARI of 0.83, indicating a high level of agreement and again supporting the validity of HiScanner's CNA calls.

## Discussion

We developed a new single-cell CNA detection algorithm, benchmarked it on real and synthetic data, and applied it to high-depth scWGS of non-diseased human brains and low-depth scWGS of longitudinal meningioma tumor samples. In our benchmarking, HiScanner achieved similar performance to a BAF-aware and BAF-agnostic single-cell CNA caller for detecting large (3–5 Mb) CNAs but outperformed them in the task of detecting small CNAs (<1 Mb) and cnLOH events. Small and cnLOH events are characteristic of many mutagenic processes and detecting them accurately is essential[41]. While our current model assumes independence between single-cell genomes for computational efficiency, future work could explore approaches such as conditional random fields or deep learning architectures to capture dependencies between adjacent genomic regions, potentially improving sensitivity in challenging regions.

In the human brain, HiScanner detected many CNAs events smaller than the broadly assumed ~3 Mb detection limit. Although the small sample size provided insufficient power to ascertain an association between CNA burden and age, HiScanner revealed previously undetected differences[12] in CNA patterns between cell types. Neuronal CNAs were predominantly private losses, in line with previous studies in human neurons[42], whereas oligodendrocytes mostly exhibited clonal gains. This finding suggests that neurons likely acquired these losses post-mitotically via non-replicative mechanisms such as non-homologous end joining and microhomology-mediated end joining, which can lead to deletions or complex genomic rearrangements without the need for DNA synthesis[43]. Indeed, neurons frequently undergo double-stranded breaks and repair due to their synaptic activity[44]. In contrast, the clonal gains found in both neurons and oligodendrocytes may have occurred earlier in brain development through replicative mechanisms such as microhomology-mediated break-induced replication and fork stalling and template switching,

which can induce duplications or deletions during DNA replication[43]. While the clonal gains in post-mitotic neurons are most likely to have occurred during development, the clonal gains in oligodendrocytes may have occurred later in life in ancestral oligodendrocyte precursor cells, which remain proliferative. Further investigation into the timing and mutational mechanisms of these CNAs with more cells is warranted. It is also important to note that while we classify CNAs as clonal when they share identical start and end bin coordinates across multiple cells, some of these events could also potentially represent independent recurrent mutations rather than true clonal expansions.

Our integrative analysis of sSNVs and CNAs provided a more detailed view of the clonal origin and evolution of a meningioma tumor, highlighting the complex and dynamic nature of cancer progression and the importance of single-cell profiling. This approach identified an ancestor of the grade III major clone in the grade II tumor, suggesting that the mutagenesis was not caused by radiotherapy. However, it remains possible that the clonal expansion of the grade III clone was influenced by selective pressure induced by radiotherapy. Timing the progression between tumor grades, and whether this occurs in response to or because of therapy, is critical information for clinical decision making.

Several studies of somatic CNAs have delineated the evolutionary trajectories of subpopulations of tumor cells, most notably in breast cancer, where chromosome-scale aneuploidy is common. A more comprehensive set of CNAs, including smaller events, will allow similar analysis for other tumor types in which the large CNAs are less common. The timing of SNV driver mutations with respect to the appearance of genome-doubling or large CNAs can be inferred based on careful analysis of variant allele fractions in bulk data[45]. With the growing availability of high-coverage scWGS and the accompanying development of methods for detecting SNVs, CNAs, and other variants, a timing analysis at the single-cell resolution will be possible, deepening our understanding of genomic stability and evolution.

## Methods
### Ethics declaration
This study was performed in accordance with all relevant ethical regulations. The meningioma study was approved by the Brigham and Women's Hospital's Institutional Review Board.

### Preprocessing of BAF
The GATK HaplotypeCaller "Best Practices" pipeline[40] with default parameters was used to call SNPs on bulk sequencing data from each individual. Germline heterozygous SNPs (gHETs) were defined as the variants passing the FILTER column flag, marked with a heterozygous "0/1" genotype, annotated non-zero population frequencies in the 1000 Genomes Project database[46] and listed in the Single Nucleotide Polymorphism Database[47]. SHAPEIT2[48] was applied to phase the gHETs using the 1000 Genomes Phase 3 integrated haplotype panel[49]. Only the gHETs that can be phased by SHAPEIT2 and have a variant allele frequency between 0.1 and 0.9 in bulk data were retained for subsequent analysis. BCFtools[50] was used to query the allelic-specific depth at each gHET site within each single-cell BAM. Next, reads associated with the same haplotypes were aggregated to obtain binwise haplotype-specific read counts. The phased B-allele frequency (pBAF) was computed as the fraction of haplotype-B read count divided by the total read count within a given bin. Note that the pBAF here corresponds to individual bins and can be heavily influenced by local allelic imbalance. To ensure robust allelic copy number inference, we implemented additional per-segment refinement of pBAF after the segmentation step using the LOH test (Supplementary Notes).

### Preprocessing of RDR
BICseq2-norm[25] with default parameters was used to perform GC correction and binning on single-cell and bulk sequencing data from

each individual. Only uniquely mappable reads with a minimum mapping quality of 30 were considered and only mappable genomic loci, as determined by the 100mer mappability track from https://www.math.pku.edu.cn/teachers/xirb/downloads/software/BICseq2/BICseq2.html, were considered.

### Allelic dropout analysis

Loss or under-amplification of one allele during whole-genome amplification, known as ADO, is a well-documented phenomenon in single-cell sequencing assays[28]. In the sequencing data, dropout regions manifest with reduced RDRs and imbalanced allelic ratios, closely resembling regions with copy number losses. The characteristics and prevalence of dropout can be affected by several factors, including amplicon length and sequencing depths. Consequently, different single-cell assays may exhibit varying patterns of dropout, and these nuances should be accounted for when determining the optimal bin size for CNA analysis in single cells.

To determine the distribution of dropout sizes without relying on arbitrary thresholds, we designed a HMM with two distinct states —"ADO" and "non-ADO"—and used Gaussian distribution probabilities to describe them (Supplementary Fig. 1). This model assumes that CNAs are infrequent compared to allelic imbalance by dropout in the selected single-cell genomes (in this study we used oligodendrocytes). We used a binary representation of BAF data at gHETs on the autosomes. A value of |BAF-0.5| less than 0.2 was marked as "0" to indicate no sign of a homozygous state, while a value of 0.2 or greater was marked as "1" to indicate a homozygous state. The HMM parameters are learned using the Baum-Welch algorithm (implemented via the hmmlearn library in Python), making the approach adaptable to any amplification method. To demonstrate this flexibility, we applied our framework to both PTA and MDA data, using oligodendrocytes with minimal aneuploidy as training examples (1278-Oligo-5 for PTA; 1278_ct_p1G9 for MDA). The trained HMMs were then used to predict dropout events across additional cells (1278-Oligo-{7,8} for PTA; 1278_ct_{p1E3, p1E6, p2B9, p2E4, p2F5, p2G5} for MDA). For each amplification method, we selected the 95th percentile of the observed dropout size distribution as the minimum bin size, allowing the framework to automatically adapt to the characteristics of different protocols.

### Joint segmentation algorithm

We extended the single-sample segmentation algorithm implemented by BICseq2-seg[25] to the multi-sample case. Within the original BIC-seq framework, the simplified BIC is defined as:

$$BIC(\lambda) = -2\sum_{j=0}^{m}\left[a_j\left(log\left(\frac{a_j}{a_j+b_j}\right)+b_j log\left(\frac{b_j}{a_j+b_j}\right)\right)\right]+(m+1)\lambda\, log(N) \tag{1}$$

where for every breakpoint $j$, $a_j$ and $b_j$ are the number of observed reads and the number of expected reads between breakpoints $\beta_j$ and $\beta_{j+1}$; $\lambda > 0$ is the tuning parameter; $m$ is the number of breakpoints; and $N = \sum_{j=0}^{m} a_j$ is the total number of reads in the given cell.

Recognizing that the joint log-likelihood of the short reads from $G$ single-cell samples is just the sum of the individual log-likelihoods, and assuming independence in cells' segmentation patterns, the multi-sample form of the simplified BIC becomes:

$$BIC(\lambda) = -2\sum_{k=1}^{G}\sum_{j=0}^{m}\left[a_{kj}\left(log\left(\frac{a_{kj}}{a_{kj}+b_{kj}}\right)+b_{kj} log\left(\frac{b_{kj}}{a_{kj}+b_{kj}}\right)\right)\right] \\ +(m+1)G\lambda\, log\left(\sum_{k=1}^{G}N_k\right) \tag{2}$$

While this formulation assumes independence between cells, which may be violated in cases where cells share clonal origins (e.g.,

tumor samples), this approximation is reasonable since (i) even in clonal populations, substantial cell-to-cell variation exists due to both technical factors (e.g., amplification noise) and biological mechanisms (e.g., acquisition of new CNAs in subclones); and (ii), modeling the full dependency structure between cells would be computationally expensive for large datasets.

Given an initial set of bins, HiScanner attempts to reduce the overall BIC by merging the neighboring bins. HiScanner calculates the BIC for each possible merge of adjacent bins. If the calculated BIC for a merged pair is lower than it would be if they remained separate, HiScanner considers these bins for merging. Out of all possible merges, HiScanner selects the pair of bins that, if merged, would result in the greatest reduction in the BIC. The selected bins are then merged. HiScanner repeats this process, continuing to merge bins pair by pair, until no more merges would result in a lower BIC.

### Ploidy and copy number inference

We extended the likelihood-based method introduced in CHISEL[19] to infer ploidy and allelic copy number states. The procedure has two main steps: identification of all possible values of the scale factor $\gamma$, and finding the maximum likelihood estimate of allelic copy number and $\gamma$ based on BIC. The scale factor $\gamma$ is a cell-specific constant value and describes the linear relationship between the total copy number and RDR for each bin. It follows that knowing the true copy number and RDR of one bin is sufficient to compute the corresponding $\gamma$. We assume that within any cell, there exists a substantial proportion of bins unaffected by CNAs, retaining balanced allelic copy numbers. The allelic copy number of these bins takes the form {1,1} when there is no whole genome duplication (WGD), {2,2} with one WGD, or {3,3} with two WGDs, etc. CHISEL's global clustering approach allows for directly selecting the largest cluster with BAF≈0.5 as the "balanced" bin set, even when these bins are not contiguous. In contrast, HiScanner's bottom-up segmentation restricts bin merging to spatially adjacent bins with similar RDR and BAF values. In HiScanner, the "balanced" bins are identified by requiring 0.45 < BAF < 0.55. HiScanner employs a Gaussian Mixture Model (GMM) to further segregate these "balanced" bins into clusters with similar RDR values. To determine the best GMM, HiScanner conducts a parameter search for the number of Gaussian components, iterating through $\Theta = \{1,...,t\}$, where $t$ denotes the maximum number of allowed WGD events. Only the set of bins in the largest cluster is chosen as the set of "balanced" bins. The identification of candidate $\gamma$ relies on these balanced bins, where both alleles have equal copy numbers. We express the set of candidate $\gamma$ values as:

$$\Gamma = \left\{\gamma = \frac{2\theta}{\frac{1}{|S|}\sum_{i\in S}x_i} : \theta\, \epsilon\, \Theta\right\} \tag{3}$$

(i.e., equation 38 in CHISEL[19]'s Supplementary Note) where $\Theta$ represents possible allele-specific copy numbers, $S$ is the set of balanced bins identified through BAF clustering, $x_i$ is the RDR of bin i, and $|S|$ is the total number of balanced bins. The numerator $2\theta$ represents the expected total copy number for balanced bins, while the denominator calculates the mean RDR across all balanced bins. This formulation systematically identifies potential scale factors that map observed read depths to absolute copy numbers, accounting for varying levels of genome duplication—from normal diploid state ($\theta = 1$, resulting in {1,1}), through single WGD ($\theta = 2$, yielding {2,2}), to multiple WGD events ($\theta = 3$ for {3,3}, and so on).

Next, HiScanner introduces a homozygosity testing framework that leverages local GC bias and allelic imbalance to derive robust per-segment BAF estimates (Supplementary Note). The segment-wise RDR is simply the mean RDR of all bins in a given segment. By modeling both RDR and BAF as Gaussian mixtures for each allelic copy number state, HiScanner applies the maximum likelihood formulation in

CHISEL[19] (Supplementary Note 5.3) to find the most probable allelic copy number state for each segment.

## Simulation of male synthetic diploid X chromosomes

We follow a procedure similar to that described in Luquette et al.[11] to construct SDX chromosomes from three single oligodendrocytes from two neurotypical male donors. For simplicity, here we denote the individuals as A and B, and the cells as OligoA, OligoB1, and OligoB2. We simulated three types of CNAs: single copy losses, single copy gains, and cnLOHs at 13 different sizes following a geometric progression from 200 kb to 5 Mb (Supplementary Data 1). For each CNA type and size, we randomly generated 100 start sites within chromosome X, after excluding pseudoautosomal regions and germline CNAs regions. For single-copy losses, we removed reads within the randomly generated start and end sites from OligoB1. For single-copy gains, we added reads mapped to CNA coordinates in OligoB2 to OligoB1. For cnLOH, we did not make changes to the read depth (only to BAF, described below), as both copies should be independently sampled in real data. After the read removal or addition, the modified OligoB1-reads and OligoA-reads were merged with SAMtools[50] to create the final SDX.

We next simulated gHETs and BAFs. To resemble a realistic SNP density, we randomly selected five positions in each 10 kb region to serve as simulated gHET loci. Only the reads uniquely mapping to the simulated gHETs are used to compute BAF. For cnLOH, only the reads from OligoA at CNA regions were used. Additional Poisson noise was added to allele-specific read depths at gHETs to represent sequencing errors. Furthermore, we simulated phase-switch errors that can occur during reference-based phasing by introducing an indicator variable $i$ to determine whether to flip the allele-specific read depths. Specifically, $i$ follows a binomial distribution with a probability of 0.01.

To evaluate performance across different sequencing depths and bin sizes, we created additional simulated datasets. For sequencing depth analysis, we downsampled our simulated 30× data to generate 10× and 0.5× coverage datasets. For bin size analysis, we tested performance using 50 KB, 100 KB, and 500 KB bins on 30× data. All combinations of depth and bin size were evaluated using the same CNA types and sizes described above, allowing systematic assessment of these parameters' effects on detection sensitivity and precision.

## Copy number calling with Ginkgo, SCOPE, SCYN, and CHISEL

Ginkgo was run with default parameters with the "binning" parameter set to "`variable_100000_101_bwa`" and hg19 genome. Ginkgo's command line version was downloaded from https://github.com/robertaboukhalil/ginkgo. For SCOPE, we used the R package to process BAM files with a 100 kb resolution, MAPQ threshold of 40, and GC/mappability correction using built-in hg19 tracks. Normalization was performed using `normalize_codex2_ns_noK` followed by `normalize_scope_foreach` with $K=1$ and $T=1:7$. Copy number segmentation was done using `segment_CBScs` with integer mode. SCYN was run with 100 kb bins using SCOPE's normalized profiles as input, performing dynamic programming-based segmentation. For the single-neuron comparison, we applied CHISEL's complete pipeline to both single-cell PTA data and paired bulk sequencing data from the same individual. This was done separately for bin sizes of 5 Mb and 100 kb. Regardless of bin size, we noticed a tendency for high ploidy predictions and overfitting, similar to what was reported previously[21]. To address this, we fine-tuned several parameters in the *chisel_calling* module to optimize the method's performance. Specifically, we set the *sensitivity* parameter to 10, *maxploidy* to 3 and *k* to 10. In addition there was a strong GC bias in CHISEL-preprocessed RDR for 100 kb bins, likely due to the lack of GC-normalization in the CHISEL pipeline. To ensure fairness in the comparison of segmentation algorithms during the synthetic diploid experiment, we used the same RDR and BAF data preprocessed by HiScanner as input for both HiScanner's and CHISEL's

segmentation algorithms. CHISEL's bioconda package was downloaded from https://github.com/raphael-group/chisel. We used bwa-mem[51] to align all the data presented in this manuscript to the "hg19/GRCh37+decoy" reference genome.

## HiScanner analysis of single neurons and oligodendrocytes

We analyzed PTA-amplified scWGS data and paired bulk whole-genome sequencing from two public datasets of eleven neurotypical brains[12,14]. Each brain includes 2–3 neurons and 2–3 oligodendrocytes whose genomes were subject to high-depth sequencing (20–30× per cell). HiScanner was run separately for each brain, using the PTA and bulk sequencing data as input and a common bin size of 100 kb, as determined by our ADO size model. Two out of 33 neurons were found to be aneuploid, displaying extensive copy number losses. No oligodendrocytes were aneuploid. Given our focus on megabase-scale mosaic CNAs, we excluded the aneuploid neurons from subsequent analyses of CNA length and prevalence. Single-cell CNA calls that overlapped with bulk CNA calls were removed. To maintain consistency across samples from both sexes, our analysis was restricted to autosomal chromosomes.

We employed a LME model framework to assess the relationship between CNA count and age across different cell types and CNA types. The statistical analyses were implemented using the *statsmodels* library in Python, using the *mixedlm* function from the *statsmodels.formula.api* module. We analyzed four combinations: gains in neurons, gains in oligodendrocytes, losses in neurons, and losses in oligodendrocytes. The dependent variable in our LME models was the CNA count (*cna_count*), with age as the predictor variable. To account for the within-individual correlation of CNA counts, we included random intercepts by specifying the individual's identifier (*indID*) as the grouping variable for random effects. This approach allowed us to model the intra-subject variability and adjust for the non-independence of the data. The formula for the LME models was set as *cna_count ~ age*, fitting each of the four specified datasets.

## Estimation of false-discovery rate (FDR)

We computed mean FDR values for each CNA size and type assessed in the synthetic diploid experiment. Each CNA detected in the single neuron and oligodendrocyte data was matched with the FDR value from the same event type (e.g., gain or loss) with the largest size smaller than the CNA event. The final FDR estimate is then calculated as the mean of FDRs across all CNAs. This produces a conservative estimate of FDR since FDR increases as CNA size decreases.

## Development of ViScanner

We developed a web-based visualization platform in order to visualize HiScanner output. It is a React-based web application hosted on GitHub. The interactive visualization is based on Higlass[27]. Custom plugin tracks, tailored to the output files of HiScanner, were created for HiGlass. When using ViScanner, the user's input files are loaded into the local browser cache and are not transferred to a server, thus enabling safe use with confidential data.

## Meningioma data generation

The samples were collected from the operation room and directly frozen in −80 °C after written informed consent was obtained under the auspices of a human subjects institutional review board protocol approved by the Brigham and Women's Hospital. The histological WHO Grade for each meningioma was determined by board-certified neuropathologists. The samples were shipped to Bioskryb, Inc. and stored at −80 °C until nuclei isolation. Nuclei were isolated using the NUC-V2 protocol on an S2 Genomics Singulator 100 machine. ResolveDNA reactions were run for 1 plate for each sample for whole-genome amplification. All cells passing quality control metrics (PTA yield) were then subjected to library

preparation and sequenced on a NextSeq 1000 instrument targeting 8 M paired-end reads at 2 × 150 bp.

**HiScanner analysis of the single-cell meningioma dataset**
HiScanner was run on the entire set of initial and recurrent meningioma scWGS as well as paired tumor and blood bulk sequencing, with a bin size of 500 kb. Hierarchical clustering was performed using R package *hclust* with complete linkage and canberra distance.

**Somatic SNV analysis of the single-cell meningioma dataset**
GATK4 Mutect2[40] with default parameters was run on tumor bulk separately for initial and recurrent samples, using the blood bulk as a control. Only variants with "PASS" flag were kept. SECEDO was run on the entire set of initial and recurrent meningioma single cells, with pileup positions restricted to Mutect2-called SSNVs.

**Reporting summary**
Further information on research design is available in the Nature Portfolio Reporting Summary linked to this article.

## Data availability
Newly generated scWGS data for 179 PTA-amplified single-cells and two matched tumor/blood bulk WGS of the initial and recurrent meningioma samples have been deposited in the European Genome-Phenome Archive (EGA) under dataset ID EGAD50000001254 under controlled access to protect the privacy of human genomics data. Researchers may request access by contacting EGA's Data Access Committee (DAC ID: EGAC50000000512, contact: mark.johnson3@umassmemorial.org). Requests are typically processed within 4-6 weeks, and approval requires a Data Use Agreement. The ground truth CNA coordinates used in the simulation experiments are provided in Supplementary Data 1. Previously generated scWGS and matched bulk WGS of PTA neurons are available for download at dbGaP, with accession number phs001485.v3.p1 [https://www.ncbi.nlm.nih.gov/projects/gap/cgi-bin/study.cgi?study_id=phs001485.v3.p1]. Access requires dbGaP approval. scWGS of MDA and PTA oligodendrocytes are available for download at NIAGADs with accession number NG00162. Access requires NIGADS DSS approval.

## Code availability
HiScanner is open-source and available on GitHub (https://github.com/parklab/HiScanner)[52]. HiScanner output can be visualized through ViScanner (https://parklab.github.io/ViScanner/).

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

## Acknowledgements

We extend our deepest gratitude to the donors and their families, whose generous contributions are pivotal to the advancement of science. This work was supported by R01HG012573 and R01CA269805 to P.J.P., F31CA264958 to V.V.V., a Sontag Foundation grant to M.D.J., K08-NS128074 to D.D.S., Grant 62587 from the John Templeton Foundation to C.A.W., and D.D.S. C.A.W. is an investigator of the Howard Hughes Medical Institute (HHMI) and supported by R01NS032457. We thank Teng Gao, Viktor Ljungström, Antuan Tran, and Dominika Maziec for their valuable input on this work.

## Author contributions

Study design: Yifan Z., L.J.L., V.V.V., P.J.P.; HiScanner algorithm and software development: Yifan Z.; simulation and caller performance benchmark: Yifan Z.; generation and interpretation of meningioma data: H.W.Y., M.D.J.; meningioma data analysis: Yifan Z.; implementation of multi-sample segmentation module: X.W., R.X.; implementation of ViScanner: A.D.V.; split-read-based validation: Yuwei Z., Yifan Z.; interpretation of non-diseased brain CNA call set: D.D.S., C.A.W.; manuscript writing: Yifan Z., P.J.P., L.J.L., with contributions from all authors.

## Competing interests

P.J.P. is a member of the scientific advisory board (SAB) for Bioskryb Genomics, Inc. C.A.W. is a member of the SAB of Bioskryb Genomics, Inc. (cash, equity), Mosaica Therapeutics (cash, equity), and an advisor to Maze Therapeutics (equity). The remaining authors declare no competing interests.
