## [Transparent Peer Review file · Nature Communications]

High-resolution Detection of Copy Number Alterations in Single Cells with HiScanner

Corresponding Author: Dr Peter Park

Version 0:

Reviewer comments:

Reviewer #1

(Remarks to the Author)

The work describes a method for improving the resolution of copy number alterations that can be detected using single cell whole genome amplification data. There is a need for higher accuracy in the single-cell copy number aberration assessment.

The abstract claims, "When applied to high-coverage scWGS data... HiScanner shows superior ability to detect smaller CNAs..." Then the paper goes on to analyze 179 cells from "longitudinal meningioma samples". First, the abstract uses the plural for cells and samples, but the 179 cells came from single patient. It would be helpful to clarify how many cells, surgical samples, and patients were involved in the study. Second, the reported depth of coverage for these real samples was 0.5x; the actual average depth of coverage is not clear and the actual depth of coverage may be quite far from the 29x used to demonstrate the method in Figure 3. The equivalent depth of coverage for the simulation data is not clear. Therefore, it is not evident whether the precision and sensitivity results in Figure 1 apply to the depth of coverage obtained in the analysis of the data in Figure 5. While the claims in the abstract do seem to be truthful, they may obscure a potential disconnect between the simulation and real data analysis. It would be helpful to supplement with low-coverage simulation results so that the precision and accuracy on the meningioma patient can be assessed.

The joint segmentation algorithm (line 522) seems to assume that both the single-cells samples are independent (the likelihood values are summed) and that the segmentation patterns are independent. Yet, in the real data analysis, the 179 single-cell samples are drawn from the same patient and therefore may be correlated. Violations of independence assumptions can lead to a dramatic overestimation of statistical significance. It would be helpful for the authors to describe why these assumptions are reasonable for the data sets they are analyzing and to what degree their methods are insensitive to violations of the independence assumption.

(Remarks on code availability)

I visited the github website hosting the HiScanner code. I was able to download and install the python package according to the instructions. However, I ran into some difficulty when trying to use the software. The "Quick Start" Section seems to contain some code examples, but the example json file used for input cannot be run because it references files that are private to the authors. The documentation section does not contain documentation except for a file on contributions and a blank page for how to prepare input. There seems to be a link and mention of a tutorial, but the link leads to an image and I was not able to identify any tutorial files in the repository. For these reasons I was not able to test the code or replicate the paper's findings. Finally, the code dependencies listed in the paper do not seem to list version information for dependent packages, so it is not clear what versions of dependent software is required.

Reviewer #2

(Remarks to the Author)

This is a very welcome development of code for analysis of single cell CNV calls after short read WGS, with application to previously published data as well as new data generated from a meningioma, the first of its kind. This field is expanding and of considerable importance, but, as the authors say, some tools are only suitable for multi-Mb CNVs, and the most widely used (Ginkgo) has not been formally updated for many years. The key additional parameter here is introduction of B allele

frequency, during the calling rather than post-hoc, and its use to optimise bin size for a given dataset. The results is indeed smaller resolution with several examples suggesting it is more accurate than Ginkgo. The method is very clearly described, and the presentation and figures of high quality. A visualisation web tool is also presented.

I believe that the manuscript can be improved further before publication, and hopefully all or at least most of these points can be addressed.

It is likely as the authors state that this pipeline is indeed correct where it is discrepant with Ginkgo, but the lack of an orthogonal gold standard for each single cell is a problem in all such studies. Using high coverage scWGS with PTA, which is said to provide more complete WGA than other methods, should allow direct visualisation of breakpoints in at least some cases. This was performed with some success for Picoplex and MDA (<https://pubmed.ncbi.nlm.nih.gov/23630320/>- not my work). It would also be extremely helpful to obtain a feel for what proportion of eg DEL and DUP SV calls in PTA scWGS PE data are supported by Hiscanner CN changes, and are not chimeras. I appreciate that not every gain or loss in scWGS data can be validated by paired end sequencing, and that the complete profile of PTA chimeras is beyond the scope of this work, but I believe that some effort could be made along these lines.

The authors make an interesting observation about clonal CNVs, which I presume means identical breakpoints in two or more cells. Were there ever more than two? Was any "wobble" allowed to determine clonality, eg only one breakpoint identical? Is there a possibility that some of these are not truly clonal but recurrent, as reported in a recent large brain scWGS study (also not my work - <https://www.nature.com/articles/s41467-024-48392-0>). In cases where CNVs do appear clonal, the authors could look for discordant read pairs / split reads in the vicinity of the CN transition regions to support these calls in high coverage bulk WGS data if available , although if the bulk coverage is only ~30x this is probably not likely to yield clear results.

The authors report "smoother read depth signals across the genome" for HiScanner, but it would be nice to quantify this. Is it possible to present MAPD or some other similar metric? Or do the authors feel that the inclusion of BAF in bin size determination makes MAPD redundant? Are the discrepancies mostly in noisy cells? eg in fig 3f the Ginkgo losses do look wrong, but the profile looks quite wavy, so perhaps these Ginkgo false calls are an issue mostly / only in noisy cells? In fig 3g, the discrepant region seems to have BAF 0 for some SNPs. Is there any significance to this?

The authors use a mix of published high coverage data and novel low coverage data. I don't get a feel for how coverage impacts data quality, which is crucial if planning CNV-only calling with WGS which as the authors says is often <1x. Is it possible to present some analyses after downsampling to, say, 10x, 5x, and 1x?

I note that Hg19 was used, possibly to be compatible with Ginkgo as originally developed. I presume Hiscanner will work on newer references, as hopefully hg19 will not be used much going forward.

The authors filter using mapq 30. This appears quite strict. I can see why it is necessary if calling SNVs, but I would feel for binning reads the criteria could be more relaxed. Would be interested to hear the authors' views, and whether they feel that different mapq could lead to significant differences.

I would like some more information on the data used, I assume it was aligned hg19 throughout, Which aligner was used for meningioma? Also I assume that bowtie2 was used in the available PTA scWGS datasets (since the Ginkgo setting for bowtie is used)? Is it possible to perform some analysis with BMA mem, which is more often used?

The authors mention determining bin size for MDA, but do not come back to this. As they know CNV calling after MDA has been problematic, but it would be nice to see how Hiscanner would perform at bins of 1 Mb or more in MDA scWGA. IS CNV calling in MDA possible at all, at least for large calls with high coverage?

Fig S3 To investigate the effect of age on CNVs, samples are divided into 4 groups, and a "weak slope" for oligos is mentioned. I would prefer the analysis to be presented as a linear regression with age as a continuous variable. If the data do not meet the requirements for this, it should be possible to present a Spearman correlation.

(Remarks on code availability)

I have not yet run the code. I do intend to do see after revision if asked to review again. Part of the reason is that I do not yet have data of this exact type now, but I do expect in in the next 1-2 months.

Reviewing the GitHub briefly,

I tried to install on mac via conda env, and it seems to work.

However, the Github link to "tutorial" leads to image of a logo

The "prepare_input.md" file has no meaningful content

Reviewer #3

(Remarks to the Author)

This paper introduces HiScanner that is based on both read counts and BAF for identifying single-cell copy number alterations from scDNA-seq data. The authors showed the higher performance of HiScanner than other two methods, i.e., Ginkgo and CHISEL, on simulated datasets. The authors also applied HiScanner to two real datasets, and the results show the effectiveness of HiScanner in profiling CNAs. Despite that it shows some novelty compared to existing methods, I have

some concerns that need to be addressed.

Major concerns:

1) the authors compared their method to Ginkgo (using read counts only) and CHISEL (relying on both read counts and BAF). Despite that Ginkgo is a popular method for calling single-cell CNAs, there are several more recently developed methods that uses read counts to infer single-cell copy number profiles, such as SCOPE, SCICoNE, SCONCE and rcCAE. These methods are built using different models (e.g., CBS, BIC, MCMC, HMM), and were reported to show comparable or better performance when compared to Ginkgo, can the authors explain the reason for choosing Ginkgo over other methods in performance comparison? Comparing the proposed HiScanner to more recent methods will enhance the contribution of this work.

2) the authors simulated three-types of CNAs, i.e., single-copy loss, single-copy gain or copy-neutral LOH. Simulation is an appropriate way to assess the methods, but I think these simple datasets are not enough to comprehensively evaluate the performance of their method. Can they explain why other more complex types of CNAs are not generated? Can the authors show if HiScanner generalizes well to datasets containing complex structure of CNAs?

3) for allelic dropout analysis, the authors only considered gHET sites with >5 reads. They should also provide the number of gHET sites after filtering with the threshold. In addition, considering the low sequencing coverage of scDNA-seq, the authors should explain why they select the threshold of >5 reads to filter gHET sites.

4) it will be good if the authors provide evaluation results of HiScanner with resolution of <100kb bin size (e.g., 50kb), such that show its advantages over existing methods, as 100kb can be seen a resolution limit of current methods.

(Remarks on code availability)

Reviewer #4

(Remarks to the Author)

HiSCANNER review comments:

The authors proposed a new CNA caller at single-cell level which is allele specific, called HiScanner. HiScanner can detect smaller CNAs due to that it has a mechanism to calculate the distribution of the ADO size such that the bin size does not have to be large. Such a calculation is based on the assumption that the CNAs are not as frequent as ADO, and thus the HMM can be used to detect ADO. Another improvement of HiSCANNER over CHISEL is that in calculating BAF, it does not use EM algorithm which is slow. Rather, it utilizes the GC content to detect LOH by shifting one allele by a few bins, and calculating so called "augmented BAF". By comparing the CDF of the mirrored augmented BAF and the original mirrored BAF, the KS test can tell whether there is a LOH event. This will further help to identify the BAF.

The authors used both simulated dataset and several real datasets to benchmark HiSCANNER, particularly compared to Ginkgo and CHISEL.

While the technique of identifying the smallest bin size to overcome the ADO, as well as escaping the EM algorithm in inferring the BAF are both very novel and intriguing, I raise a few comments below.

Major comments:

First, the training of the HMM in identifying the bin size is based on one sample by MDA and one sample by PTA. Have the authors tried the training on other type of data and other samples of MDA and PTA, and do they show the similar distribution of ADO length? In particular, DOP-PCR shall be included in the training dataset as it produces datasets suitable for CNA detection whereas MDA usually is applied to detect SNVs since the coverage that MDA generates is much higher. Further, why don't the authors just train HMM by Baum-Welch algorithm, so that the users are not limited to MDA and PTA?

Second, why doesn't HiSCANNER joint segment the genomic regions based on both RDR and BAF but only on RDR?

Third, in both the simulated and real experiments, more CNA single cell callers shall be compared in addition to Ginkgo and CHISEL. For example, SCOPE has been shown to have good performance over Ginkgo and needs to be incorporated. In addition, HMMcopy needs to be compared with too. Recently, there are a few new tools such as SeCNV, SCYN as well as DeepCNA. Authors should compare with a few of the latest tools too.

Fourth, the authors did not evaluate cnLOH on the brain data, but this is one advantage of allele-specific based method. If cnLOH is not considered, why not just use RDR-based methods?

Fifth, in diploid X chromosome simulated experiment, only single-copy gain and loss are simulated. It does not consider higher-copied gain/loss. Is HiSCANNER limited to only three states? If not, more CN states shall be simulated to make the data more realistic.

Sixth, in the brain experiment, since the authors made the hypothesis about the number of gains and losses versus the age for both neurons and oligodendrocytes, the authors should provide a figure to show such a hypothesis is valid. The authors should have all the data to show such a figure.

Seventh, in the meningioma experiment, it seems to me that the reason that the tetraploid clones in grade III (cluster 8) is inferred mainly because of the copy number 3 state regions. However, these regions do not occupy a large percentage of the genome. Also, from Fig. 5g, although from RDR there is some signal of copy number 3 state, the BAF does not show a clear evidence of deviated values, raising a question that it is possible that these copy number 3 states do not exist, and the inference is due to the noise in RDR. Since HiSCANNER is much more sensitive to the small region CNAs, it is also possible to falsely infer the CNAs at these small regions and falsely inflate the whole genome's ploidy. By the comparison with the SNV-based method (SECEDO), it seems that cluster 8 is not supported by the SNVs in Fig. 5h. Much more thorough orthogonal analysis needs to be done to show that the cells in cluster 8 are truly tetraploid. Could the authors tune the parameters in HiSCANNER? This may lead to the disappearance of cluster 8.

Minor:

A typo on line 532, $\log(b_{\{kj\}} / (a_{\{kj\}} + b_{\{kj\}}))$

In line 562, we recommend the authors write out the equation 38 in Chisel's supplemental note for the convenience of the readers, as well as cite CHISEL.

Fig. S2 doesn't have a-d labeled. Particularly, I cannot find "BAF oscillates between 0 and 1 due to imperfect phasing of gHETs (Figure S2d)". Fig. S2 is supposed to have a-f, but I can only see four panels.

In the section of "Accurate estimation of the B-allele frequency (BAF)", do the authors hold an assumption that LOH events are much larger than dropout regions? If so, please state in the text. If not, please explain why the authors aim at LOH events but are not concerned about dropout regions which may lead to the zero BAFs as well.

Fig5i is titled "SCANNER ...", while the paper proposes a method called HiSCANNER. It is unclear if SCANNER is a method compared to HiSCANNER at first glance.

(Remarks on code availability)

Reviewer #5

(Remarks to the Author)

(Remarks on code availability)

Version 1:

Reviewer comments:

Reviewer #1

(Remarks to the Author)

The authors have addressed my concerns. The claims in the abstract are better supported by the evidence. The additional simulations clarify where the proposed method is expected to perform well and where it may not. The independence assumptions are more clearly stated.

The independence assumption in the model is a limitation that is justified by the authors in part for computational reasons. It would be beneficial to the community, if space permits, to identify this assumption as a potential area for future work in the discussion. The bioinformatics and machine learning communities have abundant methods for relaxing the assumption yet maintaining computational feasibility which may improve the method in the regimes where the sensitivity is low. I would hope that the larger community would build on this work and the authors could help guide that work in their discussion.

(Remarks on code availability)

The structure of the github repository is much improved. The authors have described the operating systems that they tested

the software on. A typical pipeline of this complexity would involve building a docker or apptainer container for all of the dependencies and the github repository has sufficient detail to allow a user to construct one for their use.

Reviewer #2

(Remarks to the Author)

The authors have performed significant additional work and have addressed all my comments and concerns. I am fully supportive of this important manuscript now being published.

One aspect that perhaps could be looked into more in the future by this group and others working on single cell CNV / SV calling would be the possibility to use "high" (>20x) coverage scWGS to determine breakpoints of CNVs called. The authors make a very clear case of why this is even more challenging than I had anticipated due to chimeras, despite the good breadth of genome amplification by PTA. I wonder whether this is partly due to the strict mapping quality filtering applied, which the authors also fully justify. It would be interesting (but beyond the present study scope) to see if less stringent mapping quality filtering allows more CNV breakpoints to be defined by looking at split reads, but also discordant read pairs.

Christos Proukakis, UCL

(Remarks on code availability)

I note the updated Github. I am afraid I simply did not have the time to review it within the narrow timeframe. I do intend to test the code in due course, including in my own data, which is with different amplification method, and will be interesting to compare.

Reviewer #3

(Remarks to the Author)

The authors have addressed my concerns.

(Remarks on code availability)

Reviewer #4

(Remarks to the Author)

The authors answered all my questions. Good work.

(Remarks on code availability)

The documentation is now well done. It is great to see ViScanner's GitHub page listed as well.

Reviewer #5

(Remarks to the Author)

(Remarks on code availability)

Point-by-point responses to reviewer comments

We would like to thank the reviewers for their highly constructive comments and suggestions, which have enabled us to improve our manuscript substantially. Below we provide our responses to each comment (in blue), with edited text shown in green with corresponding line numbers.

Reviewer #1 (Remarks to the Author):

1. The work describes a method for improving the resolution of copy number alterations that can be detected using single cell whole genome amplification data. There is a need for higher accuracy in the single-cell copy number aberration assessment.

We appreciate the reviewer's acknowledgement of the importance of our method.

2. The abstract claims, "When applied to high-coverage scWGS data... HiScanner shows superior ability to detect smaller CNAs..." Then the paper goes on to analyze 179 cells from "longitudinal meningioma samples". First, the abstract uses the plural for cells and samples, but the 179 cells came from single patient. It would be helpful to clarify how many cells, surgical samples, and patients were involved in the study.

Our abstract has been revised to more precisely describe the datasets used in our study:

- For the high-coverage brain dataset, we now specify "high-coverage scWGS data from 65 cells across 11 neurotypical human brains" instead of just "human brain cells".
- For the low-coverage meningioma dataset, we now clarify "179 cells sampled from the same meningioma patient at two time points" instead of using the plural "longitudinal meningioma samples". Additionally, in our Table 3, we have listed the names, type and depth, and tumor grade. In the caption, we clearly mentioned "Tumors of both grade came from the same patient."

Our updated abstract is pasted below (lines 22-37):

“Improvements in single-cell whole-genome sequencing (scWGS) assays have enabled detailed characterization of somatic copy number alterations (CNAs) at the single-cell level. Yet, current computational methods are mostly designed for detecting chromosome-scale changes in cancer samples with low sequencing coverage. Here, we introduce HiScanner (High-resolution Single-Cell Allelic copy Number caller), which combines read depth, B-allele frequency, and haplotype phasing to identify CNAs with high resolution. In simulated data, HiScanner consistently outperforms state-of-the-art methods across various CNA types and sizes. When applied to high-coverage scWGS data from 65 cells across 11 neurotypical human brains, HiScanner shows a superior ability to detect smaller CNAs, uncovering distinct CNA patterns between neurons and oligodendrocytes. We also generated low-coverage scWGS data from 179 cells sampled from the same meningioma patient at two time points. For this serial dataset, integration of CNAs with point mutations revealed evolutionary trajectories of tumor cells. These findings show that HiScanner enables accurate characterization of frequency, clonality, and distribution of CNAs at the single-cell level in both non-neoplastic and neoplastic cells.”

3. Second, the reported depth of coverage for these real samples was 0.5x; the actual average
depth of coverage is not clear and the actual depth of coverage may be quite far from the 29x
used to demonstrate the method in Figure 3. The equivalent depth of coverage for the simulation
data is not clear. Therefore, it is not evident whether the precision and sensitivity results in
Figure 1 apply to the depth of coverage obtained in the analysis of the data in Figure 5. While the
claims in the abstract do seem to be truthful, they may obscure a potential disconnect between
the simulation and real data analysis. It would be helpful to supplement with low-coverage
simulation results so that the precision and accuracy on the meningioma patient can be assessed.

We would like to clarify that our study analyzed two distinct datasets: 1) high-coverage (20-30X)
scWGS from eleven neurotypical human brains; 2) low-coverage (0.5X) scWGS from
longitudinal meningioma pairs. In line 543, Results section titled “Integrative CNA and SNV
analysis in longitudinal meningioma scWGS”, we had described the depth of low-coverage
scWGS: “To demonstrate the applicability of HiScanner to low-coverage scWGS, we sequenced
PTA-amplified single cells from meningiomas, the most common primary intracranial tumor, to
an average depth of 0.5X (Table S3).” In line 878-880, Methods section titled “HiScanner
analysis of single neurons and oligodendrocytes”, we had described the depth of high-coverage
scWGS: “Each brain includes 2 to 3 neurons and 2 to 3 oligodendrocytes whose genomes were
subject to high-depth sequencing (20-30X per cell).” We think that the wider range of coverage
in our simulation should reduce any chance of confusion.

To address the reviewer’s concern about performance across different coverage depths, we have
expanded our benchmarking experiments to include multiple sequencing depths (0.5X, 10X, and
30X) and bin sizes (50KB, 100KB, and 500KB). Our new analysis confirms that: higher
sequencing depth (30X) provides optimal performance across all CNA types. Low-coverage
sequencing (0.5X) is robust for large CNAs (5Mb+, >0.9 sensitivity and precision) but shows
reduced performance for smaller variants, especially for those below 1Mb. These findings are
presented in the updated Supplementary Figure S3b and lines 326-373, also pasted below:

We next assessed the impact of bin size and sequencing depth on HiScanner's performance
across different CNA types. *[Omitted content about bin size benchmarking]*. Regarding
sequencing depth, we evaluated 0.5X, 10X, and 30X coverage (Fig S3b). Higher sequencing

depth (30X) consistently improved both precision and sensitivity across all CNA types, though
 the performance gain between 10X and 30X was modest. Low-coverage sequencing (0.5X)
 showed substantially lower performance, particularly for single-copy events and smaller variants,
 though it maintained reasonable accuracy for larger CNAs (>2Mb) and two-copy gains. Notably,
 the performance difference between 10X and 30X coverage was less pronounced for cnLOH
 events, suggesting that moderate coverage might be sufficient for detecting these variants. These
 results validate that the default parameter used in our benchmarking experiment (30X depth,
 100KB bins) as providing optimal performance while indicating that cost-effective alternatives
 (e.g., 10X coverage) might be suitable for specific applications.

4. The joint segmentation algorithm (line 522) seems to assume that both the single-cells samples
 are independent (the likelihood values are summed) and that the segmentation patterns are
 independent. Yet, in the real data analysis, the 179 single-cell samples are drawn from the same
 patient and therefore may be correlated. Violations of independence assumptions can lead to a
 dramatic overestimation of statistical significance. It would be helpful for the authors to describe
 why these assumptions are reasonable for the data sets they are analyzing and to what degree
 their methods are insensitive to violations of the independence assumption.

We appreciate the reviewer's thoughtful analysis of our joint segmentation algorithm's
 independence assumption. Indeed, this is an important consideration since cells from the same
 tumor often share CNV profiles due to clonality. While this violates the independence
 assumption of our joint segmentation algorithm, we argue that the independence approximation
 remains reasonable for two reasons:

- 1) Despite clonality, substantial subclonal diversity exists due to: (a) new CNAs arising in
 subclones after the initial clonal expansion; (b) technical noise and biological variability
 at the single-cell level leading to diverse readouts

2) While modeling the full dependency structure would be more accurate, it becomes
computationally too expensive for large datasets. Our independence approximation
provides a practical balance between computational feasibility and biological accuracy.
For future development, potential approaches include: (a) clustering cells into clonal groups and
treating clusters as independent units; (b) using composite likelihood methods that can account
for correlations without fully modeling them.
We have added these considerations to the Methods. Specifically, in lines 723-728, we included
the following:
“While this formulation assumes independence between cells, which may be violated in cases
where cells share clonal origins (e.g., tumor samples), this approximation is reasonable since (i)
even in clonal populations, substantial cell-to-cell variation exists due to both technical factors
(e.g., amplification noise) and biological mechanisms (e.g., acquisition of new CNAs in
subclones); and (ii) modeling the full dependency structure between cells would be
computationally expensive for large datasets.”

**Reviewer #1 (Remarks on code availability):**

I visited the github website hosting the HiScanner code. I was able to download and install the
python package according to the instructions. However, I ran into some difficulty when trying to
use the software. The "Quick Start" Section seems to contain some code examples, but the
example json file used for input cannot be run because it references files that are private to the
authors. The documentation section does not contain documentation except for a file on
contributions and a blank page for how to prepare input. There seems to be a link and mention of
a tutorial, but the link leads to an image and I was not able to identify any tutorial files in the
repository. For these reasons I was not able to test the code or replicate the paper's findings.
Finally, the code dependencies listed in the paper do not seem to list version information for
dependent packages, so it is not clear what versions of dependent software is required.

We apologize for the oversight. We have now comprehensively updated the GitHub repository to
address all the concerns raised:

- 1) Added a complete tutorial with step-by-step instructions and executable examples
- 2) Provided demo data for users to check installation. We provide three mouse single
lymphocytes (Tn5-based scWGS from Rohrbach et al¹). These were chosen because they
are publicly available on SRA whereas our data are only available through protected
access at EGA.
- 3) Created detailed documentation that cover input file preparation and requirements as well
as usage examples for different scenarios.
- 4) Specified version requirements for all dependencies in pyproject.toml.

The repository now includes all necessary resources for users to test the code and run HiScanner
on their own datasets. We invite the reviewer to visit the updated repository.

Reviewer #2 (Remarks to the Author):

This is a very welcome development of code for analysis of single cell CNV calls after short read WGS, with application to previously published data as well as new data generated from a meningioma, the first of its kind. This field is expanding and of considerable importance, but, as the authors say, some tools are only suitable for multi-Mb CNVs, and the most widely used (Ginkgo) has not been formally updated for many years. The key additional parameter here is introduction of B allele frequency, during the calling rather than post-hoc, and its use to optimise bin size for a given dataset. The results is indeed smaller resolution with several examples suggesting it is more accurate than Ginkgo. The method is very clearly described, and the presentation and figures of high quality. A visualisation web tool is also presented.

We thank the reviewer for their positive assessment of our work and for highlighting several key aspects of our tool.

I believe that the manuscript can be improved further before publication, and hopefully all or at least most of these points can be addressed.

1. It is likely as the authors state that this pipeline is indeed correct where it is discrepant with Ginkgo, but the lack of an orthogonal gold standard for each single cell is a problem in all such studies. Using high coverage scWGS with PTA, which is said to provide more complete WGA than other methods, should allow direct visualisation of breakpoints in at least some cases. This was performed with some success for Picoplex and MDA (<https://pubmed.ncbi.nlm.nih.gov/23630320/>- not my work). It would also be extremely helpful to obtain a feel for what proportion of eg DEL and DUP SV calls in PTA scWGS PE data are supported by Hiscanner CN changes, and are not chimeras. I appreciate that not every gain or loss in scWGS data can be validated by paired end sequencing, and that the complete profile of PTA chimeras is beyond the scope of this work, but I believe that some effort could be made along these lines.

We agree that an orthogonal validation using another technology would be ideal, but as the reviewer recognizes, it is not easy to do it for single cells. That is why we have relied on simulated data for performance statistics.

We did not pursue the split read approach because our previous investigation into the use of split reads for SV detection turned out to be unsuccessful. Nonetheless, we have carefully followed the reviewer's suggestion and explored the feasibility of using split and discordant reads as an orthogonal validation approach. Our preliminary analysis confirms that there are significant limitations with this approach for CNA calls in PTA-amplified single cells, detailed below.

We used CNA calls from paired meningioma tumor-normal bulk samples (identified via Battenberg²) as ground truth. First, we excluded whole-chromosome and chromosome-arm events from this analysis since telomeric and centromeric regions are highly repetitive, leading to

191 unreliable read mapping that could confound split-read detection. Second, for the remaining
192 focal CNAs, we examined split-read support at CNV boundaries, including 100Kb flanking
regions across all single cells (this 100Kb window was chosen to match the settings used in our
high-depth brain cell analysis: 20-30X coverage, 100Kb bin size). A CNA candidate was
considered validated only if split reads supported both breakpoints. When pooling all
meningioma single cells (n=179, ~90X equivalent total depth), only 8 out of 40 CNAs (20%)
showed split-read support at both breakpoint ends. At 25X depth (50 cells), the mean validation
rate dropped to 15.67% (SD=2%), indicating that even true CNAs rarely exhibit split-read
support in single cells given our coverage and resolution. Furthermore, PTA amplification
generates numerous chimeric DNA molecules, producing abundant split reads throughout the
genome. Using all chimeric reads without requiring support at both breakpoint ends would
artificially validate nearly every CNA call. This creates a paradox where the validation approach
is either too stringent (requiring both ends) or too permissive (accepting any chimeric read).
Given these limitations, we conclude that split-read validation cannot reliably assess the
accuracy of CNA calls in PTA-amplified single cells. Instead, we rely on our extensive
benchmarking using simulated data, where ground truth is known, to derive FDR estimates as is
described in Methods (lines 906-912), pasted below:

**Estimation of false-discovery rate (FDR)**

We computed mean FDR values for each CNA size and type assessed in the synthetic diploid
experiment. Each CNA detected in the single neuron and oligodendrocyte data was matched with
the FDR value from the same event type (e.g., gain or loss) with the largest size smaller than the
CNA event. The final FDR estimate is then calculated as the mean of FDRs across all CNAs.
This produces a conservative estimate of FDR since FDR increases as CNA size decreases.

2. The authors make an interesting observation about clonal CNVs, which I presume means
identical breakpoints in two or more cells. Were there ever more than two? Was any "wobble"
allowed to determine clonality, eg only one breakpoint identical? Is there a possibility that some
of these are not truly clonal but recurrent, as reported in a recent large brain scWGS study (also
not my work - <https://www.nature.com/articles/s41467-024-48392-0>). In cases where CNVs do
appear clonal, the authors could look for discordant read pairs / split reads in the vicinity of the
CN transition regions to support these calls in high coverage bulk WGS data if available ,
although if the bulk coverage is only ~30x this is probably not likely to yield clear results.

The reviewer is correct to assume that "clonal CNV" in our manuscript means identical
breakpoints in two or more cells. As discussed in our response to point #1 above, the moderate
coverage (~30X) of our bulk samples combined with the inherent challenges of split-read
detection at CNV boundaries makes split-read-based CNV validation approach unreliable for
evaluating these events. Instead, we relied on the consistency of both read depth and B-allele
frequency signals across multiple cells within each individual to support the clonal nature of
these CNVs. We now characterize the distribution of clonal CNVs and include discuss this
phenomenon in our Results section, lines 481-483:

We identified 72 clonal events (defined as CNAs sharing identical start and end bin coordinates
across multiple cells), with the majority (50/72) being shared between more than two cells,
demonstrating substantial clonal diversity.

Regarding breakpoint determination: Our analysis pipeline uses joint segmentation, which
enforces consistent breakpoint positions across all cells from the same individual. We require
both breakpoints to match exactly and do not allow "wobble" in breakpoint positions. We
acknowledge that some events we classify as clonal could potentially be independent recurrent
events, as recently shown in the study the reviewer cites. Definitively distinguishing between
these scenarios would require ultra-high-depth bulk sequencing data, which is beyond the scope
of our current study. In light of this, we included the following in the Discussion (lines 628-631):

It is also important to note that while we classify CNAs as clonal when they share identical start
and end bin coordinates across multiple cells, some of these events could also potentially
represent independent recurrent mutations rather than true clonal expansions.

3. The authors report "smoother read depth signals across the genome" for HiScanner, but it
would be nice to quantify this. Is it possible to present MAPD or some other similar metric? Or
do the authors feel that the inclusion of BAF in bin size determination makes MAPD redundant?
Are the discrepancies mostly in noisy cells? eg in fig 3f the Ginkgo losses do look wrong, but the
profile looks quite wavy, so perhaps these Ginkgo false calls are an issue mostly / only in noisy
cells?

We thank the reviewer for this insightful suggestion. The MAPD metrics for the high-coverage
brain cells were previously reported in the original
publications (Luquette et al, Nature Genetics 2022, and
Ganz et al Cell 2024). However, to address this specific
concern: we plotted the MAPD distribution of the high-
coverage PTA single neurons and oligodendrocytes
used in this paper (see figure on the left), which shows
that single neurons and oligodendrocytes amplified by
PTA exhibit MAPD values of 0.52 ± 0.05 and 0.55 ± 0.03
respectively, compared to bulk samples (0.25 ± 0.05).
The neuron presented in Figure 3 (5823PFC-B) has an
MAPD of 0.5575, which falls within the typical range
of PTA-amplified neurons. This indicates that the false
calls by Ginkgo are not due to unusual noise levels in
this particular cell, but rather represent a general limitation in handling PTA-amplified samples
without BAF information.

4. In fig 3g, the discrepant region seems to have BAF 0 for some SNPs. Is there any significance
to this?

We thank the reviewer for this astute observation.
 While one bin (100kb) in the discrepant region
 (highlighted in salmon shade in the figure on the left)
 shows BAF=0, our statistical analysis suggests these
 are more likely to be allelic dropout events rather than
 true deletions. From our detailed dropout analysis
 (Figure S1), we established that PTA cells sequenced
 at 20-30X coverage typically show dropout events of
 approximately 1000 base pairs, which matches the bin
 size used for BAF and RDR estimates in this figure.
 Importantly, the zero BAF values in this region are not
 consecutive but rather scattered, a pattern more
 consistent with random dropout events than with a
 genuine deletion. Additionally, other methods tested
 on this region (SCYN and SCOPE) agree with
 HiScanner's interpretation, providing further support
 for our conclusion.

5. The authors use a mix of published high coverage
 data and novel low coverage data. I don't get a feel for how coverage impacts data quality, which
 is crucial if planning CNV-only calling with WGS which as the authors says is often <1x. Is it
 possible to present some analyses after downsampling to, say, 10x, 5x, and 1x?

We thank the reviewer for raising an important point about sequencing depth's impact on CNA
 detection performance. We have comprehensively addressed this in our response to Reviewer #1,
 Point #3, where we present new benchmarking results across multiple sequencing depths (0.5X,
 10X, and 30X). Our analysis reveals a key insight: while the performance difference between
 0.5X and 10X coverage is substantial, the improvement from 10X to 30X is relatively modest.
 This observation suggests that moderate coverage (10X) may be sufficient for many applications,
 providing practical guidance for experimental design in CNV-only studies. Please refer to our
 detailed response above and the updated Supplementary Figure S3b (also pasted below), which
 shows precision and sensitivity measurements for each sequencing depth and CNA type. We also
 included the following in main text (lines 326-373):

We next assessed the impact of bin size and sequencing depth on HiScanner's performance
 across different CNA types. *[Omitted content about bin size benchmarking]*. Regarding
 sequencing depth, we evaluated 0.5X, 10X, and 30X coverage (Fig S3b). Higher sequencing
 depth (30X) consistently improved both precision and sensitivity across all CNA types, though
 the performance gain between 10X and 30X was modest. Low-coverage sequencing (0.5X)
 showed significantly reduced performance, particularly for single-copy events and smaller
 variants, though it maintained reasonable accuracy for larger CNAs (>2Mb) and two-copy gains.
 Notably, the performance difference between 10X and 30X coverage was less pronounced for
 cnLOH events, suggesting that moderate coverage might be sufficient for detecting these
 variants. These results validate that the default parameter used in our benchmarking experiment

(30X depth, 100KB bins) as providing optimal performance while indicating that cost-effective
 alternatives (e.g., 10X coverage) might be suitable for specific applications.

6. I note that Hg19 was used, possibly to be compatible with Ginkgo as originally developed. I
 presume Hiscanner will work on newer references, as hopefully hg19 will not be used much
 going forward.

We have updated the GitHub repository with comprehensive documentation and resources to
 support HiScanner analysis using both hg19 and hg38 reference genomes. The repository now
 includes detailed instructions and reference files needed for hg38-based analyses.

7. The authors filter using mapq 30. This appears quite strict. I can see why it is necessary if
 calling SNVs, but I would feel for binning reads the criteria could be more relaxed. Would be
 interested to hear the authors' views, and whether they feel that different mapq could lead to
 significant differences.

We appreciate the reviewer's thoughtful comment about our MAPQ threshold. A MAPQ of 30
 corresponds to a 0.1% probability of incorrect mapping, meaning we have 99.9% confidence in
 the read placement. While this threshold may appear strict, it is particularly important in single-
 cell sequencing where whole-genome amplification can generate chimeric DNA molecules -
 artificial fusions of fragments from different genomic regions. These chimeric reads often
 receive lower MAPQ scores due to their ambiguous mapping. By requiring $\text{MAPQ} \geq 30$, we
 effectively filter out these problematic reads that could introduce noise in copy number
 estimation. While a more relaxed threshold would increase the number of usable reads, the
 presence of chimeric reads and other mapping artifacts could potentially impact the accuracy of
 copy number detection. We believe that MAPQ 30 threshold represents a conservative choice
 that prioritizes mapping quality over coverage depth.

8. I would like some more information on the data used, I assume it was aligned hg19 throughout, Which aligner was used for meningioma? Also I assume that bowtie2 was used in the available PTA scWGS datasets (since the Ginkgo setting for bowtie is used)? Is it possible to perform some analysis with BMA mem, which is more often used?

We apologize for any confusion regarding the alignment details. All data in this manuscript was aligned to the "hg19/GRCh37+decoy" reference genome using BWA-MEM (we now clarify this detail in line 873). In our initial analysis, we incorrectly used Ginkgo's bowtie-based binning setting. We have now rerun the Ginkgo analysis using settings that properly match our BWA-MEM alignment approach (--genome hg19 --binning variable_100000_101_bwa, line 850) and have updated all results and figures accordingly. The changes in Ginkgo settings did not materially affect our comparative analyses or conclusions.

9. The authors mention determining bin size for MDA, but do not come back to this. As they know CNV calling after MDA has been problematic, but it would be nice to see how Hiscanner would perform at bins of 1 Mb or more in MDA scWGA. IS CNV calling in MDA possible at all, at least for large calls with high coverage?

We appreciate the reviewer's interest in MDA-amplified data. We have been interested in amplicon sizes in MDA data (e.g., Sherman et al, PaSD-qc: quality control for single cell whole-genome sequencing data using power spectral density estimation, *Nucleic Acids Research*, 2018) to see what analysis might be possible. While we did analyze the allelic dropout patterns in MDA data to determine optimal bin sizes, we found that CNV calling in MDA-amplified single cells is generally not feasible, even at large bin sizes of 1Mb or more. Our analysis demonstrates this through a comparison of allelic balance distributions across different amplification methods and sequencing depths. In MDA data (50X coverage), we observe an extreme U-shaped distribution of allelic ratios, indicating that most genomic regions show nearly complete amplification of one allele (ratios near 0 or 1) with minimal balanced regions (ratios near 0.5). This pattern contrasts sharply with PTA-amplified cells at both high (20X) and low (0.5X) coverage, where the allelic ratios maintain a more balanced distribution centered around 0.5. The severe allelic imbalance inherent to MDA amplification makes it impossible to reliably distinguish true copy number changes from amplification artifacts, regardless of sequencing depth or bin size selection. This represents a fundamental limitation of using MDA-amplified data for CNV detection.

10. Fig S3 To investigate the effect of age on CNVs, samples are divided into 4 groups, and a "weak slope" for oligos is mentioned. I would prefer the analysis to be presented as a linear regression with age as a continuous variable. If the data do not meet the requirements for this, it should be possible to present a Spearman correlation.

We thank the reviewer for this suggestion and have now implemented a more rigorous statistical approach, analyzing age as a continuous variable through linear mixed effects (LME) models. This approach accounts for within-individual correlation of CNA counts while examining the relationship between age and CNA frequency. We have updated the manuscript with a new Figure S4 (pasted below) showing the continuous relationship between age and CNA counts, replacing the previous grouped analysis. The Methods section now includes detailed description of the statistical methodology using linear mixed effects models. The Results section now includes the following (lines 483-504):

Analysis of age-related patterns using linear mixed effects models revealed a weak negative correlation between age and non-clonal CNA gains in oligodendrocytes (Fig S4a; slope=-0.020 CNA/year, $p=0.010$). No significant age-related patterns were observed for neuronal gains (Fig S4a; slope=0.015 CNA/year, $p=0.218$) or for non-clonal losses in either cell type (Fig S4b; neurons: slope=0.003 CNA/year, $p=0.976$; oligodendrocytes: slope=0.002 CNA/year, $p=0.589$), suggesting that the relationship between CNAs and aging may be more complex than described in prior literature¹⁵.

Reviewer #2 (Remarks on code availability):

1. I have not yet run the code. I do intend to do see after revision if asked to review again. Part of
the reason is that I do not yet have data of this exact type now, but I do expect in in the next 1-2
419 months.

Reviewing the GitHub briefly,

I tried to install on mac via conda env, and it seems to work.

However, the Github link to "tutorial" leads to image of a logo

The "prepare_input.md" file has no meaningful content

We apologize for our oversight in having a more finished version of our repository. We have
updated the repository to include a fully functional tutorial with step-by-step instructions and
executable examples. The previous broken tutorial link has been corrected. The
prepare_input.md file has been replaced with a comprehensive README.md.

While we understand the reviewer has not yet run the code due to data constraints, we encourage
revisiting the repository once the necessary data is available. We also provide a demo dataset for
users to test their installation.

**Reviewer #3 (Remarks to the Author):**

This paper introduces HiScanner that is based on both read counts and BAF for identifying
single-cell copy number alterations from scDNA-seq data. The authors showed the higher
performance of HiScanner than other two methods, i.e., Ginkgo and CHISEL, on simulated
datasets. The authors also applied HiScanner to two real datasets, and the results show the
effectiveness of HiScanner in profiling CNAs. Despite that it shows some novelty compared to
existing methods, I have some concerns that need to be addressed.

1. The authors compared their method to Ginkgo (using read counts only) and CHISEL (relying
on both read counts and BAF). Despite that Ginkgo is a popular method for calling single-cell
CNAs, there are several more recently developed methods that uses read counts to infer single-
cell copy number profiles, such as SCOPE, SCICoNE, SCONCE and rcCAE. These methods are
built using different models (e.g., CBS, BIC, MCMC, HMM), and were reported to show
comparable or better performance when compared to Ginkgo, can the authors explain the reason
for choosing Ginkgo over other methods in performance comparison? Comparing the proposed
HiScanner to more recent methods will enhance the contribution of this work.

We thank the reviewer for this suggestion to expand our method comparisons. We had chosen
Ginkgo because it was simple but popular method and CHISEL because of its sophisticated
methodology, but we agree that a more comprehensive comparison would be of benefit. We have
now included two additional recent methods - SCOPE and SCYN. SCOPE employs a Poisson
latent factor model for bias correction and copy number estimation, while SCYN leverages
SCOPE's normalization framework and implements dynamic programming for efficient
segmentation.

We carefully considered but ultimately excluded several other methods that the reviewer listed
for technical reasons:

- SCONCE requires matched normal single cells, making it unsuitable for our evaluation framework which focuses on detecting private CNVs in individual cells
- SCICoNE has not yet been peer-reviewed as it is currently only available as a preprint
- Methods like rcCAE and other deep learning approaches are designed for datasets with large numbers of cells and are not optimized for detecting private CNVs in individual cells, which is the focus of our benchmarking

The two additional algorithms have been incorporated throughout the Results (see lines 232-324)
and Methods (see lines 848-874), particularly in our simulation studies and evaluation using real
single-neuron data. The expanded benchmarking demonstrates HiScanner's superior performance
across various CNA types and sizes, while providing a more comprehensive assessment of the
current methodological landscape in single-cell CNA detection (see Figures 2 and 3). We note
that a recent benchmarking study (Mallory et al., PLOS Comp Bio 2020) provides detailed
comparisons between CBS-based and HMM-based approaches, showing that CBS-based
methods like Ginkgo generally achieve better performance.

2. The authors simulated three-types of CNAs, i.e., single-copy loss, single-copy gain or copy-
neutral LOH. Simulation is an appropriate way to assess the methods, but I think these simple
datasets are not enough to comprehensively evaluate the performance of their method. Can they

explain why other more complex types of CNAs are not generated? Can the authors show if
HiScanner generalizes well to datasets containing complex structure of CNAs?

It is not clear what the reviewer has in mind for the more complex CNAs, but given the difficulty
of working split reads in PTA data, it seems unlikely that we can learn anything other than the
simple copy number status that may have resulted from complex SVs and CNAs. Nonetheless,
we have expanded our evaluation to include two-copy gains in addition to our original set of
CNA types. This addition proved valuable, as it revealed important differences in how various
tools handle higher amplitude copy number changes.

Analysis of two-copy gains (Fig 2f, g) revealed an interesting limitation of certain algorithmic
approaches. While HiScanner and Ginkgo demonstrated nearly identical high performance for
these higher amplitude events above 1Mb (~0.85 sensitivity, ~0.90 precision at 2.5Mb+),
SCOPE and SCYN were not able to detect these changes at all.

After investigating SCOPE's normalized read depth profiles, we found that the underlying issue
stems from its Poisson latent factor normalization approach. SCOPE uses an EM algorithm that
aims to distinguish technical biases from biological signal. However, when presented with high-
amplitude changes like two-copy gains, its normalization algorithm appears to mistakenly
interpret these as technical artifacts rather than true copy number changes. This is rooted in
SCOPE's core design - it was optimized to detect subtle copy number changes while aggressively
correcting for technical noise. When analyzing a small number of cells, SCOPE's normalization
over-corrects these stronger signals. This explanation is consistent with the original SCOPE
paper's benchmarking, which focused primarily on single-copy alterations. Similarly, SCYN
inherits this limitation since it uses SCOPE's normalized profiles as input. The other tools like
HiScanner and Ginkgo employ different normalization strategies that are more tolerant of high-
amplitude changes, allowing them to successfully detect two-copy gains while still maintaining
high precision.

While even more complex CNA patterns exist in cancer genomes (such as high-level
amplifications or complex rearrangements), our current evaluation focuses on copy number
changes that can be reliably detected from read depth and BAF signals. Complex rearrangements
typically require split-read evidence for accurate characterization, which is challenging to obtain
from single-cell data due to their highly chimeric nature from whole genome amplification (see
our response points #1-2 to Reviewer #2). We believe that our current evaluation spanning
single-copy losses/gains, two-copy gains, and copy-neutral LOH provides a strong foundation
for assessing method performance across the most commonly encountered CNA types that can
be detected from coverage-based signals in both normal and cancer cells.

3. for allelic dropout analysis, the authors only considered gHET sites with >5 reads. They should also provide the number of gHET sites after filtering with the threshold. In addition, considering the low sequencing coverage of scDNA-seq, the authors should explain why they select the threshold of >5 reads to filter gHET sites.

In our initial analysis, we applied a >5 read threshold for high-quality gHET selection. However, upon further consideration, we recognized that this threshold was not critical, as reads covering gHETs were aggregated across parental haplotypes. To validate this, we performed allelic dropout analyses with and without the >5 read threshold and observed no difference in optimal bin sizes that were predicted. Based on these results, we have removed the >5 read requirement and updated our analysis by removing the sentence “Only gHET sites with >5 reads were considered.”

4. It will be good if the authors provide evaluation results of HiScanner with resolution of <100kb bin size (e.g., 50kb), such that show its advantages over existing methods, as 100kb can be seen a resolution limit of current methods.

We thank the reviewer for their suggestion to evaluate HiScanner’s performance at bin sizes below 100kb, particularly at 50kb, to demonstrate its advantages over existing methods. As highlighted in the updated main text (lines 326-332), we have conducted a comprehensive analysis of HiScanner’s performance across multiple bin sizes (50kb, 100kb, and 500kb) using simulated 30X data (Supplementary Figure S3a, pasted below).

We also included the following in lines 326-332 to describe HiScanner's performance across bin sizes:

We next assessed the impact of bin size and sequencing depth on HiScanner's performance across different CNA types. For bin size analysis, we compared 50KB, 100KB, and 500KB bins in simulated 30X data (Fig S3a). While smaller bins (50KB) showed improved sensitivity for detecting variants below 500kb, they generally exhibited lower precision due to increased noise. The 100KB bin size provided optimal balance between sensitivity and precision across most CNA types, particularly for variants >1Mb. Larger bins (500KB) demonstrated reduced performance, especially for smaller variants.

**Reviewer #4 (Remarks to the Author):**

HiSCANNER review comments:

The authors proposed a new CNA caller at single-cell level which is allele specific, called
HiScanner. HiScanner can detect smaller CNAs due to that it has a mechanism to calculate the
distribution of the ADO size such that the bin size does not have to be large. Such a calculation is
based on the assumption that the CNAs are not as frequent as ADO, and thus the HMM can be
used to detect ADO. Another improvement of HiSCANNER over CHISEL is that in calculating
BAF, it does not use EM algorithm which is slow. Rather, it utilizes the GC content to detect
LOH by shifting one allele by a few bins, and calculating so called “augmented BAF”. By
comparing the CDF of the mirrored augmented BAF and the original mirrored BAF, the KS test
can tell whether there is a LOH event. This will further help to identify the BAF.

The authors used both simulated dataset and several real datasets to benchmark HiSCANNER,
particularly compared to Ginkgo and CHISEL.

While the technique of identifying the smallest bin size to overcome the ADO, as well as
escaping the EM algorithm in inferring the BAF are both very novel and intriguing, I raise a few
comments below.

Major comments:

1. First, the training of the HMM in identifying the bin size is based on one sample by MDA and
one sample by PTA. Have the authors tried the training on other type of data and other samples
of MDA and PTA, and do they show the similar distribution of ADO length? In particular, DOP-
PCR shall be included in the training dataset as it produces datasets suitable for CNA detection
whereas MDA usually is applied to detect SNVs since the coverage that MDA generates is much
higher. Further, why don't the authors just train HMM by Baum-Welch algorithm, so that the
users are not limited to MDA and PTA?

We thank the reviewer for their thoughtful questions about HMM training and performance
across different scWGS protocols. Regarding the reviewer's question about HMM training, our
HMM implementation already uses the Baum-Welch algorithm for parameter estimation, as it is
the default training method in the hmmlearn library that we employ. As a result, we find
consistent dropout length distributions no matter which sample we choose as training data.

In the Method (“Allelic dropout analysis”), we now include the following in line 698: “The
HMM parameters are learned using the Baum-Welch algorithm (implemented via the hmmlearn
library in Python), making the approach adaptable to any amplification method.”

To explore our pipeline's broader applicability, we expanded our analysis to include two
additional sequencing conditions that mirror characteristics of different amplification protocols:

1. Low-coverage PTA meningioma cells (0.5X), approximating the depth typical of DOP-
PCR protocols

2. Ultra-low coverage Tn5-based single neurons (0.05X, *from unpublished collaborative*
*work*), representing an even sparser coverage regime with different amplification
chemistry

Our analysis revealed an interesting relationship between coverage depth and dropout
characteristics. As shown in the figure below, in low-coverage PTA data (0.5X), we observed
longer dropout lengths compared to high-coverage PTA (20X), suggesting the need for larger bin
sizes to achieve reliable CNA detection at lower depths. However, when comparing low-
coverage PTA (0.5X) to ultra-low coverage Tn5-based data (0.05X), we found similar optimal
bin sizes despite their 10-fold difference in coverage. This suggests that once coverage drops
below a certain threshold, the dropout length distribution stabilizes. This finding has important
practical implications, as it indicates that our bin size optimization approach remains robust even
in very low coverage regimes, regardless of the specific amplification chemistry used.

While we have not directly tested DOP-PCR data as the reviewer suggested, our method's
demonstrated ability to handle ultra-low coverage data with different amplification chemistries
suggests it should work effectively with DOP-PCR protocols, which typically operate in similar
coverage ranges. The technology-agnostic nature of our approach stems from two key factors:
first, our use of the Baum-Welch algorithm for HMM parameter estimation, which we have now
clarified in the Methods section, and second, our focus on characterizing fundamental properties
of allelic dropout (i.e., allelic imbalance) rather than protocol-specific features. These elements
allow our method to adapt to various amplification methods while maintaining reliable dropout
detection and appropriate bin size selection. While these results provide important validation of
our method's broad applicability, we unfortunately cannot include the actual data visualization as
it contains preliminary results that has not yet been published.

2. Second, why doesn't HiSCANNER joint segment the genomic regions based on both RDR
and BAF but only on RDR?

We thank the reviewer for this insightful question about joint segmentation. While joint
segmentation of RDR and BAF is appealing, our extensive testing revealed significant practical
challenges that led us to favor RDR-based initial segmentation. As demonstrated in the figure
below, attempts at joint segmentation produce two major issues:

First, the inherent variance structure differs between RDR and BAF signals. While we can
effectively normalize RDR variance through GC-correction and variable binning, BAF variance
shows complex, region-specific patterns that resist simple normalization. When incorporated into
joint segmentation, this variable BAF dispersion frequently causes over-segmentation of the
genome.

Second, our empirical testing with different penalty parameters (λ) shows that joint
segmentation paradoxically reduces accuracy. As visible in panels c-e of the figure below,
increasing the penalty parameter to control over-segmentation leads to missing true breakpoints
(orange arrows). This creates an unfortunate trade-off between sensitivity and specificity that
proves difficult to optimize.

To address these challenges, HiScanner instead uses a two-step approach: initial segmentation
based on RDR, followed by BAF-based refinement of the identified segments. This strategy
provides more stable results while still leveraging the valuable information content in both
signals. We acknowledge this may occasionally miss copy-neutral events that lack RDR signals,
but we provide tools through our ViScanner platform (<https://parklab.github.io/ViScanner/>) for
interactive exploration and refinement of such regions.

3. Third, in both the simulated and real experiments, more CNA single cell callers shall be compared in addition to Ginkgo and CHISEL. For example, SCOPE has been shown to have good performance over Ginkgo and needs to be incorporated. In addition, HMMcopy needs to be compared with too. Recently, there are a few new tools such as SeCNV, SCYN as well as DeepCNA. Authors should compare with a few of the latest tools too.

We thank the reviewer for raising this point about expanding method comparison. As we detailed in our response to Reviewer #3 Comment #1, we have now expanded our benchmarking to include SCOPE and SCYN in addition to Ginkgo and CHISEL. Please refer to that response for a comprehensive explanation of our tool selection criteria and the benchmarking results.

Specifically, we have edited Results (lines 232-324), Methods (lines 848-874), Figures 2 and 3 to reflect our expanded benchmarking results, now including SCOPE and SCYN as the reviewer suggested.

We also carefully considered HMMcopy, Deep CNV, SeCNV but we opted to exclude for the
following reasons:

- - HMMcopy was not included as it was primarily designed for bulk sequencing data, not
single-cell analysis. A recent benchmarking study (Mallory et al., PLOS Comp Bio 2020)
has demonstrated that CBS-based methods like Ginkgo generally achieve better
performance than HMM-based approaches in single-cell applications.
- - For DeepCNA and other deep-learning-based approaches like rcCAE, while these are
valuable tools, they are specifically optimized for large-scale datasets and employ deep
learning approaches that require substantial training data. Our work focuses on detecting
private CNVs in individual cells or small cell populations, making these tools less
suitable for our evaluation framework.
- - SeCNV constructs a Distance Correlation Matrix using a local Gaussian kernel to capture
similarities between bins across multiple cells ($n \geq 5$). This matrix is crucial for the
subsequent segmentation step, but it cannot be meaningfully constructed from a single
cell's data. This limitation is confirmed by one of our co-authors who contributed to the
development of Consequently, SeCNV is less effective for our focus on private CNVs in
individual cells or small subpopulations.

4. Fourth, the authors did not evaluate cnLOH on the brain data, but this is one advantage of
allele-specific based method. If cnLOH is not considered, why not just use RDR-based methods?

We acknowledge the importance of analyzing copy-neutral loss of heterozygosity (cnLOH)
events, as they can provide unique insights not captured by read-depth based methods. In our
initial analysis, we excluded cnLOH calls to maintain a low false discovery rate, given the lower
precision observed in our simulation experiments for small cnLOH events. However, based on
the reviewer's suggestion, we have now included cnLOH analysis but with a conservative size
threshold of 2Mb, where our simulations showed high precision and sensitivity (~0.83 for both
metrics).

In lines 521-527, we now added that “For cnLOHs, we focused our analysis on events larger than
2Mb where our simulations demonstrated high precision and sensitivity (~0.9). This analysis
identified cnLOH events in 8 neurons, with one neuron (5817PFC-B) showing an extensive
pattern of 12 distinct cnLOH events while the remaining neurons harbored 1-3 events each. Two
of these cnLOH events were clonal, appearing in pairs of neurons from the same individual,
while the rest were private. Notably, no cnLOH events were detected in oligodendrocytes,
similar to the cell-type specific pattern observed with single copy losses.”

5. Fifth, in diploid X chromosome simulated experiment, only single-copy gain and loss are
simulated. It does not consider higher-copied gain/loss. Is HiSCANNER limited to only three
states? If not, more CN states shall be simulated to make the data more realistic.

We thank the reviewer for this important question about HiScanner's ability to handle higher
copy number states. HiScanner is not limited to three states -- it can detect a broader range of
copy number states as demonstrated by our analysis of the grade III meningioma cells, where we
successfully identified copy number states ranging from 2 to 6 (Figure 5h).

As detailed in our response to Reviewer #3 Comment #2, we have expanded our simulation
framework to include two-copy gains in addition to single-copy changes. Our evaluation showed
that HiScanner maintains high performance for these higher amplitude events (sensitivity ~0.85,
precision ~0.90 for variants >1Mb). This robust performance for two-copy gains, combined with
our results from the meningioma analysis, demonstrates HiScanner's ability to detect and
accurately quantify higher copy number states.

While it is possible to simulate even higher copy states, we focused our systematic
benchmarking on the most frequently observed copy number alterations in normal and cancer
cells to ensure our evaluation reflects common real-world scenarios. This approach allows for
rigorous assessment of method performance while maintaining biological relevance. For a
detailed discussion of how HiScanner compares with other tools in higher copy number states,
please see our response to Reviewer #3 Comment 2, which elaborates on the technical
considerations and limitations.

6. Sixth, in the brain experiment, since the authors made the hypothesis about the number of
gains and losses versus the age for both neurons and oligodendrocytes, the authors should
provide a figure to show such a hypothesis is valid. The authors should have all the data to show
such a figure.

We thank the reviewer for requesting validation of our hypothesis regarding CNA frequency and
age relationships. As demonstrated in our response to Reviewer #2 Comment #10, we have
conducted a rigorous statistical analysis of this relationship using linear mixed effects (LME)
models. The data and analysis have been visualized in the updated Figure S4 (pasted below),
which clearly shows the continuous relationships between age and CNA counts for both gains
and losses across cell types. The Results section now includes the following (lines 483-504):

“Analysis of age-related patterns using linear mixed effects models revealed a weak negative
correlation between age and non-clonal CNA gains in oligodendrocytes (Fig S4a; slope=-0.020
CNA/year, p=0.010). No significant age-related patterns were observed for neuronal gains (Fig
S4a; slope=-0.015 CNA/year, p=0.218) or for non-clonal losses in either cell type (Fig S4b;
neurons: slope=0.003 CNA/year, p=0.976; oligodendrocytes: slope=0.002 CNA/year, p=0.589),
suggesting that the relationship between CNAs and aging may be more complex than described
in prior literature¹⁵.”

7. Seventh, in the meningioma experiment, it seems to me that the reason that the tetraploid
clones in grade III (cluster 8) is inferred mainly because of the copy number 3 state regions.
However, these regions do not occupy a large percentage of the genome. Also, from Fig. 5g,
although from RDR there is some signal of copy number 3 state, the BAF does not show a clear
evidence of deviated values, raising a question that it is possible that these copy number 3 states
do not exist, and the inference is due to the noise in RDR. Since HiSCANNER is much more
sensitive to the small region CNAs, it is also possible to falsely infer the CNAs at these small
regions and falsely inflate the whole genome's ploidy. By the comparison with the SNV-based
method (SECEDO), it seems that cluster 8 is not supported by the SNVs in Fig. 5h. Much more
thorough orthogonal analysis needs to be done to show that the cells in cluster 8 are truly
tetraploid. Could the authors tune the parameters in HiSCANNER? This may lead to the
disappearance of cluster 8.

We thank the reviewer for this careful analysis regarding the tetraploid state inference in cluster
8. While the reviewer is correct that copy number 3 (CN3) regions do not occupy a large

percentage of the genome, we respectfully disagree with the assessment that BAF evidence is
 lacking. We have tested multiple parameter settings in HiScanner, including stricter
 segmentation criteria and different ploidy priors, and find the tetraploid inference remained
 stable across these parameter variations, suggesting it is not an artifact of particular parameter
 choices. Through careful analysis of these regions, we find strong support for the CN3 state from
 both read depth and BAF signals. To demonstrate this more clearly, we show a zoomed in
 genome track showing read depth and mirrored BAF (i.e., $\min(\text{BAF}, 1-\text{BAF})$) patterns in the
 CN3 regions compared to flanking region. As shown in the figure, when examining a
 representative CN3 region at higher resolution, we observe a significant difference in BAF
 distributions between the CN3 segment (red shaded region) and the adjacent CN4 segment (blue
 shaded region) with a t-test p-value of $2.13e-04$. We believe this clear BAF separation, combined
 with the corresponding read depth signal changes, provides strong evidence for the existence of
 these CN3 regions.

 To facilitate transparent evaluation of such complex copy number patterns, we developed
 ViScanner, an interactive visualization tool that allows detailed examination of the underlying
 signals.

Regarding the discrepancy with SECEDO's SNV-based clustering, we had actually addressed
 this in our manuscript (line 590): "Notably, however, SECEDO's clustering did not distinguish
 diploid and tetraploid grade III cells. A possible explanation for this is that the whole-genome
 duplication preceded the majority of sSNV events."

We acknowledge that definitive ploidy validation would require orthogonal experimental
 approaches, such as DAPI-based DNA content measurement or fluorescence in situ hybridization
 (FISH). While such validation would be valuable, it is beyond the scope of the current study.

Minor:

8. A typo on line 532, $\log(b_{\{kj\}} / (a_{\{kj\}} + b_{\{kj\}}))$

We thank the reviewer for catching this typo. We have corrected the mathematical notation to read: $\log(b_{kj}/(a_{kj} + b_{kj}))$. The full formulation is pasted below (line 713)

$$BIC(\lambda) = -2 \sum_{j=0}^m [a_j \log(\frac{a_j}{a_j + b_j}) + b_j \log(\frac{b_j}{a_j + b_j})] + (m + 1)\lambda \log(N)$$

9. In line 562, we recommend the authors write out the equation 38 in Chisel's supplemental note for the convenience of the readers, as well as cite CHISEL.

Following the reviewer's suggestion, we have now included the complete equation from CHISEL's supplementary note (equation 38) in our manuscript for readers' reference. We have also added the appropriate citation to CHISEL, lines 783-799, pasted below:

To determine the best GMM, HiScanner conducts a parameter search for the number of Gaussian components, iterating through $\Theta = \{1, \dots, t\}$, where t denotes the maximum number of allowed WGD events. Only the set of bins in the largest cluster is chosen as the set of "balanced" bins. The identification of candidate γ relies on these balanced bins, where both alleles have equal copy numbers. We express the set of candidate γ values as:

$$\Gamma = \{\gamma = \frac{2\theta}{\frac{1}{|S|} \sum_{i \in S} x_i} : \theta \in \Theta\}$$

(i.e., Equation 38 in CHISEL 20's supplementary note) where Θ represents possible allele-specific copy numbers, S is the set of balanced bins identified through BAF clustering, x_i is the RDR of bin i , and $|S|$ is the total number of balanced bins. The numerator 2θ represents the expected total copy number for balanced bins, while the denominator calculates the mean RDR across all balanced bins. This formulation systematically identifies potential scale factors that map observed read depths to absolute copy numbers, accounting for varying levels of genome duplication - from normal diploid state ($\theta=1$, resulting in $\{1,1\}$), through single WGD ($\theta=2$, yielding $\{2,2\}$), to multiple WGD events ($\theta=3$ for $\{3,3\}$, and so on).

10. Fig. S2 doesn't have a-d labeled. Particularly, I cannot find "BAF oscillates between 0 and 1 due to imperfect phasing of gHETs (Figure S2d)". Fig. S2 is supposed to have a-f, but I can only see four panels.

We appreciate the reviewer pointing out this oversight. We have now added clear alphabetical
labels (a-f) to all panels in Figure S2.

11. In the section of “Accurate estimation of the B-allele frequency (BAF)”, do the authors hold
an assumption that LOH events are much larger than dropout regions? If so, please state in the
text. If not, please explain why the authors aim at LOH events but are not concerned about
dropout regions which may lead to the zero BAFs as well.

Yes, our method does assume that LOH events are typically larger than dropout regions. This
assumption is built into our analysis pipeline by design - we first determine the optimal bin size
based on the distribution of dropout events (specifically using the 95th percentile of dropout
sizes as the minimum bin size), and then perform LOH detection at this binning resolution. This
hierarchical approach allows us to distinguish between dropouts, which appear as isolated bins
with zero BAF, and true LOH events which manifest as contiguous regions of extreme BAF
values. Since dropout events are largely filtered out during the initial binning step, any remaining
segments showing consistent deviation in BAF are more likely to represent genuine LOH events
rather than technical artifacts. We have clarified this important assumption by adding the
following to the supplemental note: "Importantly, we assume that true LOH events typically span
regions larger than most allelic dropout event (i.e., the optimal bin size inferred from the data)."

However, we acknowledge that due to the inherent nature of single-cell whole genome
sequencing and amplification, it is impossible to definitively distinguish between technical
dropouts and true LOH events, particularly when the event is private to a single cell. Our method
makes a best effort to separate these signals based on their size distributions and contiguity
patterns.

12. Fig5i is titled “SCANNER ...”, while the paper proposes a method called HiSCANNER. It is
unclear if SCANNER is a method compared to HiSCANNER at first glance.

We apologize for the oversight. We have now resolved the discrepancies in the naming due to
versioning. All instances of “SCANNER” in the Figures have been renamed as “HiScanner”.

**Reviewer #5 (Remarks to the Author):**

I co-reviewed this manuscript with one of the reviewers who provided the listed reports. This is
part of the Nature Communications initiative to facilitate training in peer review and to provide
appropriate recognition for Early Career Researchers who co-review manuscripts.

We thank both reviewers for their valuable feedback. We appreciate Nature Communications'
initiative to involve and recognize Early Career Researchers in the peer review process.

**References:**

1. Rohrback, S. et al. Submegabase copy number variations arise during cerebral cortical
neurogenesis as revealed by single-cell whole-genome sequencing. Proc Natl Acad Sci U S A
115, (2018).

2. Nik-Zainal, S. et al. The life history of 21 breast cancers. Cell 149, 994–1007 (2012).

Point-by-point responses to reviewer comments

We sincerely thank all reviewers for their thoughtful comments and valuable suggestions. We are pleased that our revised manuscript has addressed the major concerns raised in the initial review process. Below, we provide specific responses to the remaining comments from each reviewer.

REVIEWERS' COMMENTS

Reviewer #1 (Remarks to the Author):

The authors have addressed my concerns. The claims in the abstract are better supported by the evidence. The additional simulations clarify where the proposed method is expected to perform well and where it may not. The independence assumptions are more clearly stated.

The independence assumption in the model is a limitation that is justified by the authors in part for computational reasons. It would be beneficial to the community, if space permits, to identify this assumption as a potential area for future work in the discussion. The bioinformatics and machine learning communities have abundant methods for relaxing the assumption yet maintaining computational feasibility which may improve the method in the regimes where the sensitivity is low. I would hope that the larger community would build on this work and the authors could help guide that work in their discussion.

We thank the reviewer for this insightful suggestion. We have added the following to the Discussion section highlighting the independence assumption as a potential area for future improvement (lines 500-503):

"While our current model assumes independence between single-cell genomes for computational efficiency, future work could explore approaches such as conditional random fields or deep learning architectures to capture dependencies between adjacent genomic regions, potentially improving sensitivity in challenging regions."

Reviewer #1 (Remarks on code availability):

The structure of the github repository is much improved. The authors have described the operating systems that they tested the software on. A typical pipeline of this complexity would involve building a docker or apptainer container for all of the dependencies and the github repository has sufficient detail to allow a user to construct one for their use.

We appreciate the positive feedback on the improved structure of our GitHub repository.

Reviewer #2 (Remarks to the Author):

The authors have performed significant additional work and have addressed all my comments
and concerns. I am fully supportive of this important manuscript now being published.

One aspect that perhaps could be looked into more in the future by this group and others working
on single cell CNV / SV calling would be the possibility to use "high" (>20x) coverage scWGS
to determine breakpoints of CNVs called. The authors make a very clear case of why this is even
more challenging than I had anticipated due to chimeras, despite the good breadth of genome
amplification by PTA. I wonder whether this is partly due to the strict mapping quality filtering
applied, which the authors also fully justify. It would be interesting (but beyond the present study
scope) to see if less stringent mapping quality filtering allows more CNV breakpoints to be
defined by looking at split reads, but also discordant read pairs.

Christos Proukakis, UCL

We thank the reviewer for this excellent suggestion. We fully agree that exploring higher
coverage scWGS (>20x) for precise CNV breakpoint determination represents an intriguing
direction for future research. The calibrated relaxation of mapping quality filtering, combined
with analysis of split reads and discordant read pairs, could potentially enable more accurate
breakpoint identification. As the reviewer notes, this is beyond the scope of the present study, but
we believe it represents an important avenue for future exploration.

Reviewer #2 (Remarks on code availability):

I note the updated Github. I am afraid I simply did not have the time to review it within the
narrow timeframe. I do intend to test the code in due course, including in my own data, which is
with different amplification method, and will be interesting to compare.

We also appreciate the reviewer's intention to test our code on data generated with different
amplification methods and look forward to any feedback or collaborative opportunities that may
arise from this comparison.

Reviewer #3 (Remarks to the Author):

The authors have addressed my concerns.

Reviewer #4 (Remarks to the Author):

The authors answered all my questions. Good work.

Reviewer #4 (Remarks on code availability):

The documentation is now well done. It is great to see ViScanner's GitHub page listed as well.

Reviewer #5 (Remarks to the Author):

I co-reviewed this manuscript with one of the reviewers who provided the listed reports. This is
part of the Nature Communications initiative to facilitate training in peer review and to provide
appropriate recognition for Early Career Researchers who co-review manuscripts.

We thank Reviewers #3, #4, and #5 for their positive feedback and for confirming that our
revisions have addressed their previous concerns. We are particularly pleased that Reviewer #4
found our improved documentation and GitHub page satisfactory.

We also extend our appreciation to the Early Career Researcher who co-reviewed our manuscript
as part of Nature Communications' peer review training initiative.